American Society for Microbiology | Microbiology Spectrum

# Thiamine-Mediated Microbial Interaction between Auxotrophic *Rhodococcus ruber* ZM07 and Prototrophic Cooperators in the Tetrahydrofuran-Degrading Microbial Community H-1

Hui Huang,[a,b] Hao Wu,[a] Minbo Qi,[a] Haixia Wang,[a] Zhenmei Lu[a]

[a]MOE Laboratory of Biosystem Homeostasis and Protection, College of Life Sciences, Zhejiang University, Hangzhou, China
[b]Institute of Translational Medicine, Zhejiang University, Hangzhou, China

**ABSTRACT** As a crucial growth factor, thiamine can regulate functional microbial communities; however, our current understanding of its effect on bioremediation is lacking. Using metatranscriptome and 16S rRNA gene sequencing, we explored the mechanism of response of an efficient tetrahydrofuran (THF)-degrading microbial culture, designated H-1, to exogenous thiamine. *Rhodococcus ruber* ZM07, a strain performing the THF degradation function in H-1, is a thiamine-auxotrophic bacterium. Furthermore, thiamine affected the microbial community structure of H-1 by altering resource and niche distributions. A microbial co-occurrence network was constructed to help us identify and isolate the cooperators of strain ZM07 in the microbial community. Based on the prediction of the network, two non-THF-degrading bacteria, *Hydrogenophaga intermedia* ZM11 and *Pigmentiphaga daeguensis* ZM12, were isolated. Our results suggest that strain ZM11 is a good cooperator of ZM07, and it might be more competitive than other cooperators (e.g., ZM12) in cocultured systems. Additionally, two dominant strains in our microbial culture displayed a "seesaw" pattern, and they showed completely different responses to exogenous thiamine. The growth of the THF degrader ZM07 was spurred by additional thiamine (with an increased relative abundance and significant upregulation of most metabolic pathways), while the growth of the cooperator ZM11 was obviously suppressed under the same circumstances. This relationship was the opposite without thiamine addition. Our study reveals that exogenous thiamine can affect the interaction patterns between THF- and non-THF-degrading microorganisms and provides new insight into the effects of micronutrients on the environmental microbial community.

**IMPORTANCE** Auxotrophic microorganisms play important roles in the biodegradation of pollutants in nature. Exploring the interspecies relationship between auxotrophic THF-degrading bacteria and other microbes is helpful for the efficient utilization of auxotrophic functional microorganisms. Herein, the thiamine-auxotrophic THF-degrading bacterium ZM07 was isolated from the microbial culture H-1, and the effect of thiamine on the structure of H-1 during THF bioremediation was studied. Thiamine may help ZM07 occupy more niches and utilize more resources, thus improving THF degradation efficiency. This research provides a new strategy to improve the THF or other xenobiotic compound biodegradation performance of auxotrophic functional microorganisms/microbial communities by artificially adding special micronutrients. Additionally, the "seesaw" relationship between the thiamine-auxotrophic strain ZM07 and its prototrophic cooperator ZM11 during THF bioremediation could be changed by exogenous thiamine. This study reveals the effect of micronutrients on microbial interactions and provides an effective way to regulate the pollutant biodegradation efficiency of microbial communities.

**KEYWORDS** tetrahydrofuran degradation, thiamine, *Rhodococcus ruber* ZM07, *Hydrogenophaga intermedia* ZM11, 16S rRNA gene sequencing, metatranscriptome

Address correspondence to Zhenmei Lu, lzhenmei@zju.edu.cn.

The authors declare no conflict of interest.

Thiamine (vitamin $B_1$) is a heterocyclic compound containing a thiazole ring and a pyrimidine ring. Thiamine pyrophosphate (TPP), the activated form of thiamine, is an important cofactor for some key enzymes in metabolism and participates in important life activities, such as fatty acid and protein metabolism [1]. In addition, TPP-dependent enzymes, such as pyruvate dehydrogenase [2, 3], acetolactate synthase [4, 5], transketolase [6], and $\alpha$-ketoglutarate dehydrogenase [7], are crucial for continuing cell growth and resisting metabolic stress.

Mammals, including humans, can obtain thiamine from food, while in contrast, most microorganisms and plants tend to have their own pathways to synthesize thiamine [8]. Interestingly, a number of microorganisms cannot synthesize thiamine *de novo* and rely on external thiamine released by other microorganisms for growth [9–11]. However, it is not clear why there are so many thiamine-auxotrophic microorganisms in nature. Previous studies considered thiamine-mediated ecological interactions that influence the survival and abundance of members of communities [12]. Thiamine supplementation increased the abundance of rumen bacteria associated with thiamine metabolism in dairy cows [13]. However, the relative abundance of auxotrophic species in gut communities was observed to be unaltered in the presence of B vitamins in great excess in the diet [14]. Therefore, exploring the effects of exogenous thiamine on both auxotrophic and prototrophic microbes and their response to thiamine disturbances, could promote understanding of thiamine's function in diverse microbial communities.

As it is an essential factor for organisms, the effects of thiamine on fermentation [15], livestock production [16, 17], and illness treatment [18, 19] have been extensively studied, but there have been few related studies in the field of environmental pollution remediation. To our knowledge, thiamine has been proven to be capable of improving the growth and degradation efficiency of a dioxane-degrading microbe, *Rhodococcus ruber* strain 219 [20], which illustrated that adding such essential growth factors can improve the process of microbial remediation. Tetrahydrofuran (THF; a structural analog of dioxane) is a universal solvent widely applied in the chemical and pharmaceutical industries. THF is easily transferred to surface water, groundwater, and the atmosphere due to its high water solubility and high vapor pressure [21]. According to toxicity tests, THF shows central nervous system toxicity, with headache, dizziness, and loss of sense of smell [22]. Environmental pollution with THF has become an extremely acute problem considering its wide application, physical properties, and toxicity. Biodegradation is an environmentally friendly and effective strategy to remove pollutants [23]. In our previous work, a highly efficient THF-degrading microbial culture, H-1, was enriched from activated sludge. We demonstrated that H-1 exhibited high THF-degrading efficiency and high THF tolerance. Additionally, microbial culture H-1 showed excellent THF degradation performance under generally harsh environmental conditions, indicating that H-1 might be a valuable material for sewage treatment [24]. Additionally, a thiamine-auxotrophic *Rhodococcus ruber* strain, ZM07, was isolated and proven to be a dominant THF-degrading bacterium in H-1 [25]. Therefore, the microbial culture H-1 is good material for investigating the effects of essential factors on auxotrophic bacteria and their cooperators in functional microbial communities.

In this study, we investigated the community structure changes and responses of the THF-degrading microbial culture H-1 to exogenous thiamine during the THF degradation process, especially the mechanism of interaction between the thiamine-auxotrophic THF-degrading bacterium and other prototrophic microorganisms in the community, using high-throughput sequencing. This research aimed to reveal the role of thiamine in the process of collaborative THF degradation by thiamine-auxotrophic bacteria and other prototrophic microorganisms.

## RESULTS

**Exogenous thiamine affected H-1 degradation of THF during transfer.** The THF-degrading microbial culture H-1 was transferred 40 times with or without exogenous thiamine. The results showed that regardless of the THF degradation ability or growth ability of H-1, it varied between the group receiving additional thiamine (the THI

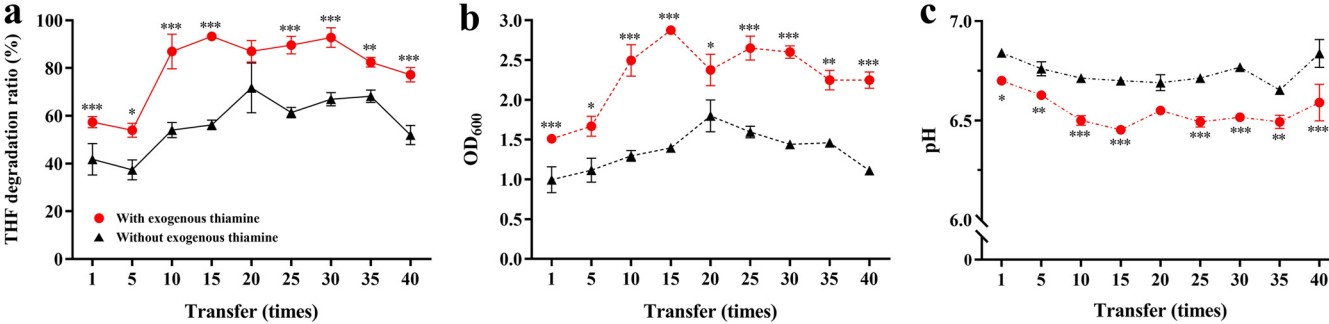

**FIG 1** Changes in the THF degradation ratio (a), biomass (b), and pH (c) curves of the enrichment culture H-1 within 48 h of cultivation every 5 transfers in 40 serial subcultures. The red line indicates the THI group (the group with exogenous thiamine), and the black line indicates the CK group (the group without exogenous thiamine). The $P$ value indicates statistical significance between the THI group and CK group determined using Student's $t$ test ($n = 3$). *, $P < 0.05$; **, $P < 0.01$; ***, $P < 0.001$. Error bars represent standard errors of the means and may be smaller than the symbol.

group, which includes T1-THI and T40-THI, i.e., samples of the 1st and 40th transfers, respectively) and the control group (the CK group, which includes T1-CK and T40-CK, i.e., samples of the 1st and 40th transfers, respectively) (Fig. 1). After 10 transfers, the THF degradation ratio remained in a narrow range of 85 to 95% in the THI group and was only 50% to 75% in the CK group (Fig. 1a), and the average total biomass reached optical densities at 600 nm ($OD_{600}$) of 2.50 in the THI group and just 1.44 in the CK group after 2 days of cultivation (Fig. 1b). The final pH values were also significantly ($P < 0.001$) affected by exogenous thiamine (Fig. 1c), which might be related to the accumulation of acidic metabolites during THF degradation. Our results suggested that exogenous thiamine could improve the performance of H-1. Compared to the culture under thiamine-poor conditions (CK group), H-1 can degrade THF more efficiently and produce more biomass under thiamine-rich conditions (THI group). The final degradation ratio and biomass of the THI group were 1.48 times ($P < 0.001$) and 2.03 times ($P < 0.001$) those of the CK group, respectively (Fig. 1a and b). Additionally, the final pH value of the THI group was just 0.96 times ($P < 0.001$) that of the CK group (Fig. 1c), probably caused by the production and accumulation of higher concentrations of acidic intermediates after degrading larger amounts of THF, as previously reported (26).

**The microbial structure was affected by exogenous thiamine.** The $\alpha$-diversity indices based on the operational taxonomic units (OTUs) showed that the bacterial community structure was affected by both exogenous thiamine and transfer. During the process from the 1st transfer to the 40th transfer, the Margalef index increased in the CK group (T40-CK versus T1-CK) while it decreased in the THI group (T40-THI versus T1-THI) (Fig. 2a; also, see Fig. S1a in the supplemental material). The Shannon evenness and Simpson index were significantly lower in T40-THI than in T1-THI, while there was no difference between T40-CK and T1-CK (Fig. 2b; Fig. S1b). These results indicated that thiamine addition affected the bacterial community diversity. According to $\alpha$-diversity analysis, there was no significant difference in community richness (Fig. 2a) and community evenness (Fig. 2b) between T1-THI and T1-CK, but the THF degradation ability and the total biomass of H-1 were significantly increased after addition of exogenous thiamine in the 1st transfer (Fig. 1). Briefly, the results showed that both the community richness and $\alpha$-diversity of T40-THI were significantly lower than those of the other three groups (T1-THI, T1-CK, and T40-CK) (Fig. 2a and b; Fig. S1), indicating that exogenous thiamine and transfer might inhibit some bacteria in microbial community H-1.

For the $\beta$-diversity index analysis, the influence of exogenous thiamine on the bacterial community was determined by nonmetric multidimensional scaling (NMDS) and principal-coordinate analysis (PCoA). Both NMDS and PCoA displayed clear separations of all groups (Fig. 2c), indicating that exogenous thiamine and transfer affected the bacterial community structure at the OTU level. Additionally, the $\beta$-diversity index

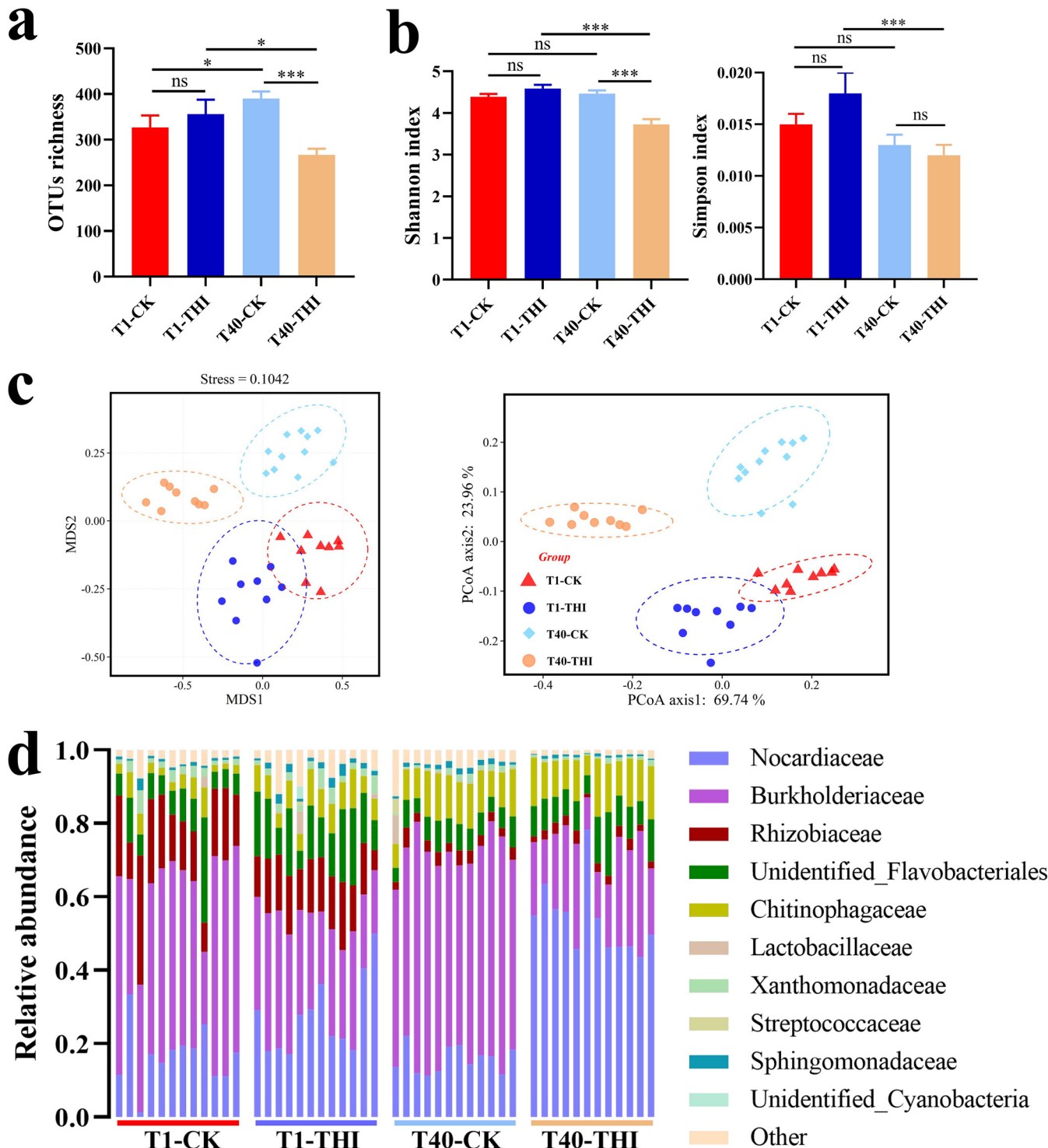

**FIG 2** The richness (a) and Shannon and Simpson indices (b) of OTUs show the $\alpha$-diversity of the microbial community. The $P$ value indicates statistical significance determined using Student's $t$ test ($n = 12$). ns, not significant ($P > 0.05$); *, $P < 0.05$; ***, $P < 0.001$. Error bars represent standard errors of the means. (c) NMDS and PCoA analyses show the $\beta$-diversity of the microbial community. The significance of clustering was determined using analysis of similarities. The dotted circles represent 95% confidence intervals. (d) Histogram of samples showing microbial composition profiling at the family level. T1-THI, with 0.01 mM thiamine at the 1st transfer; T1-CK, without thiamine at the 1st transfer; T40-THI, with 0.01 mM thiamine at the 40th transfer; T40-CK, without thiamine at the 40th transfer.

analysis also showed that the bacterial community structure was significantly shaped by both exogenous thiamine ($P < 0.001$; $F = 32.22$) and transfer ($P < 0.001$; $F = 20.37$). Therefore, we speculated that exogenous thiamine has a greater impact on the composition of the microbial community structure than the transfer process from the 1st

to the 40th transfer. The bacterial composition of each sample at the family and genus levels is shown in the bar chart (Fig. 2d; Fig. S2). In all groups, the two families with the highest relative abundance are *Nocardiaceae* and *Burkholderiaceae*, while the two genera with the highest relative abundance are *Rhodococcus* and *Hydrogenophaga* (Fig. 2d; Fig. S2). These results demonstrated that thiamine significantly affected the species diversity and community composition, but the two dominant genera did not change.

**Gene expression responses of H-1 to exogenous thiamine.** Metatranscriptome sequencing and analysis of H-1 were performed to explore the differences at the gene expression level. Kyoto Encyclopedia of Genes and Genomes (KEGG) enrichment analysis indicated that the vital differentially expressed genes (DEGs) are mainly associated with metabolism, such as carbohydrate metabolism, amino acid metabolism, and energy metabolism (Fig. S3). A large number of DEGs related to the pathways of "carbon metabolism," "ribosome biosynthesis," and "amino acid biosynthesis" were upregulated by exogenous thiamine, while DEGs related to the pathways of "ABC transporters," "two-component system," and "purine metabolism" were downregulated in microbial culture H-1 (Fig. S4). These results indicated that exogenous thiamine significantly changed the expression of genes involved in cell metabolism, biological transport, and other life activities in microbial community H-1.

The expression profile of the genes involved in thiamine metabolism can reflect the influence of exogenous thiamine on the thiamine turnover of THF-degrading and -nondegrading bacteria in the microbial community. *Rhodococcus ruber* ZM07, a thiamine-auxotrophic THF-degrading bacterium, is the only strain in the genus *Rhodococcus* in microbial community H-1 (25). In this study, *thiL*, which encodes a 35-kDa protein with thiamine monophosphate kinase activity to synthesize TPP (27), was significantly upregulated after adding exogenous thiamine to ZM07 (Table S3). However, other genes involved in thiamine biosynthesis and transport pathways in other bacteria except ZM07 were downregulated after addition of exogenous thiamine. These genera include *Rhizobium*, *Pseudaminobacter*, *Chryseobacterium*, *Chelatococcus*, *Nitratireductor*, *Bordetella*, *Proteus*, *Achromobacter*, *Ochrobactrum*, *Mesorhizobium*, *Pseudomonas*, *Devosia*, etc. Based on these results, we provide an overview of DEGs involved in thiamine metabolism between thiamine-auxotrophic ZM07 and other bacteria except ZM07 when H-1 was surrounded by sufficient thiamine in the environment (Fig. 3; Table S3). Overall, exogenous thiamine enhanced TPP synthesis by strain ZM07 but inhibited thiamine synthesis and transport in other bacteria.

Our results suggested significant differences in microbial gene expression between the THI and CK groups after 40 transfers (Fig. 4; Fig. S5). KEGG enrichment analysis revealed that carbohydrate metabolism, amino acid metabolism, energy metabolism, and nucleotide metabolism were significantly regulated by exogenous thiamine (Fig. 4a). Compared with other bacteria, most of the DEGs in ZM07 were upregulated with exogenous thiamine in H-1 (Fig. 4b). Notably, the gene cluster *thm*, encoding THF monooxygenase, which is responsible for the first step of THF hydroxylation, was significantly upregulated in ZM07 with exogenous thiamine. Meanwhile, exogenous thiamine also resulted in the upregulation of genes involved in the pathways producing coenzyme NADH and cytochrome *c* (one possible electron acceptor of Thm) (28) (Fig. 4b; Table S4). KEGG enrichment analysis revealed that the genes involved in basal metabolism, including DNA replication and repair, transcription, translation, the tricarboxylic acid cycle, and the pentose phosphate pathway, were also upregulated in ZM07 compared with the CK group (Fig. 4b). However, other bacteria performed conversely, except for some genes involved in amino acid metabolism and the electron transport chain (ETC). Otherwise, a heat map based on fragments per kilobase of transcript per millions fragments sequenced (FPKM) of DEGs revealed that hierarchical clustering grouped the samples into two clusters: the original stage (T1-CK and T1-THI) and the final stage (T40-CK and T40-THI) (Fig. S6). Most genes in the T40-THI group had lower expression than those in the other three groups. In summary, exogenous thiamine resulted in the upregulation of genes in strain ZM07 but the downregulation of genes in other bacteria in H-1, thereby improving the THF degradation efficiency of the community.

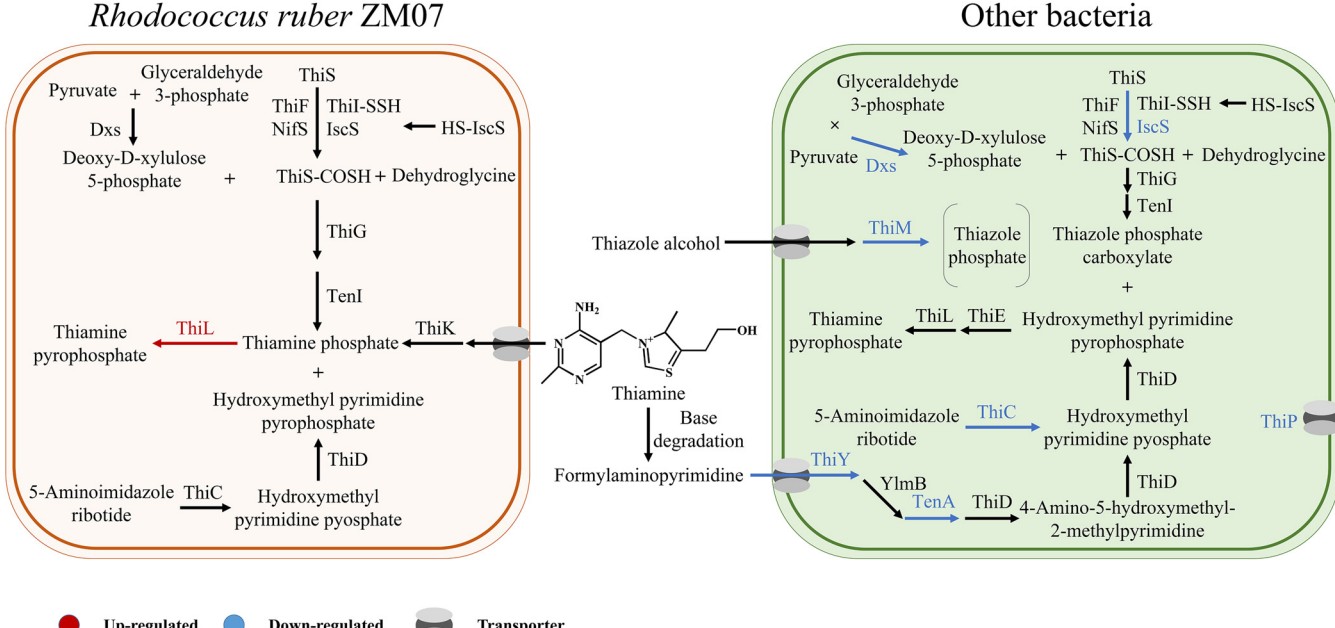

**FIG 3** Overview of differentially expressed genes involved in thiamine metabolism between thiamine-auxotrophic ZM07 and other bacteria except ZM07 (T1-THI versus T1-CK). Red text and lines represent upregulated thiamine metabolism, and blue text and lines represent downregulated metabolism. (*dxs* encodes Dxs, which catalyzes the first step in thiamine biosynthesis) (63); *iscS* encodes IscS, which transfers sulfur to ThiI in thiamine biosynthesis (64); *thiC* encodes ThiC, which converts 5-aminoimidazole ribotide to 4-amino-5-hydroxymethyl-2-methylpyrimidine phosphate (65); *thiM* encodes ThiM, a thiazole kinase (66); *tenA* encodes TenA, a thiaminase II (67); *thiY* encodes ThiY, a binding protein of the ThiXYZ transport system (68); and *thiP* encodes ThiP, an ABC-type transmembrane domain protein (69).

**Network-directed efficient isolation of bacteria related to strain ZM07.** As a thiamine-auxotrophic THF-degrading bacterium, strain ZM07 cannot grow with THF as the sole carbon source. To obtain the bacteria related to ZM07 in H-1, a microbial co-occurrence network was constructed based on 16S rRNA gene high-throughput sequencing data from the 1st and 40th transfers. According to the prediction results, only *Hydrogenophaga* and *Pigmentiphaga* had a tight correlation with ZM07 (Fig. 5a). These two target bacteria, *Hydrogenophaga intermedia* ZM11 and *Pigmentiphaga daeguensis* ZM12, were successfully isolated from H-1 and verified as nondegrading bacteria. According to the coculture experiments, both strains could help the degrader restore growth and THF-degrading ability; however, ZM11 appears to be a better cooperator of ZM07 (Fig. 5b). The coculture system of ZM07 and ZM12 requires almost 132 h to degrade 20 mM THF, while the coculture system of ZM07 and ZM11 requires only 84 h to consume the same amount of THF. The relative abundances of ZM11 and ZM12 were similar in the two-strain cocultured systems (containing strains ZM07 and ZM11 or strains ZM07 and ZM12), but ZM11 outcompeted and grew much better than ZM12 in the three-strain system (ZM07, ZM11, and ZM12) (Fig. 5c; Fig. S7a). In addition, the relative fitness of strain ZM11 was significantly higher than that of ZM12 in the three-strain cocultured system (Fig. 5d; Fig. S7b), indicating that ZM11 is a very strong competitor that can drastically suppress ZM12, precluding the coexistence of both species.

**Thiamine impacted the interactions between strain ZM07 and the cooperators.** In microbial community H-1, the proportions of strains ZM07 and ZM11 remained high in all groups, and the relative abundance of these two strains varied in an opposite way. Here, we propose a "seesaw" model to exhibit the relationship between ZM07 and ZM11 (Fig. 6a). The proportion of ZM07 in the groups with exogenous thiamine was significantly ($P < 0.05$) higher than that in the groups without thiamine (T1-THI versus T1-CK and T40-THI versus T40-CK), and ZM11 showed the opposite (Fig. 6b and Fig. S8). According to Spearman's correlation analysis, strain ZM11 had the strongest correlation with the THF-degrading strain ZM07 (Fig. 6c). The coculture of ZM07 and ZM11 achieved the best THF

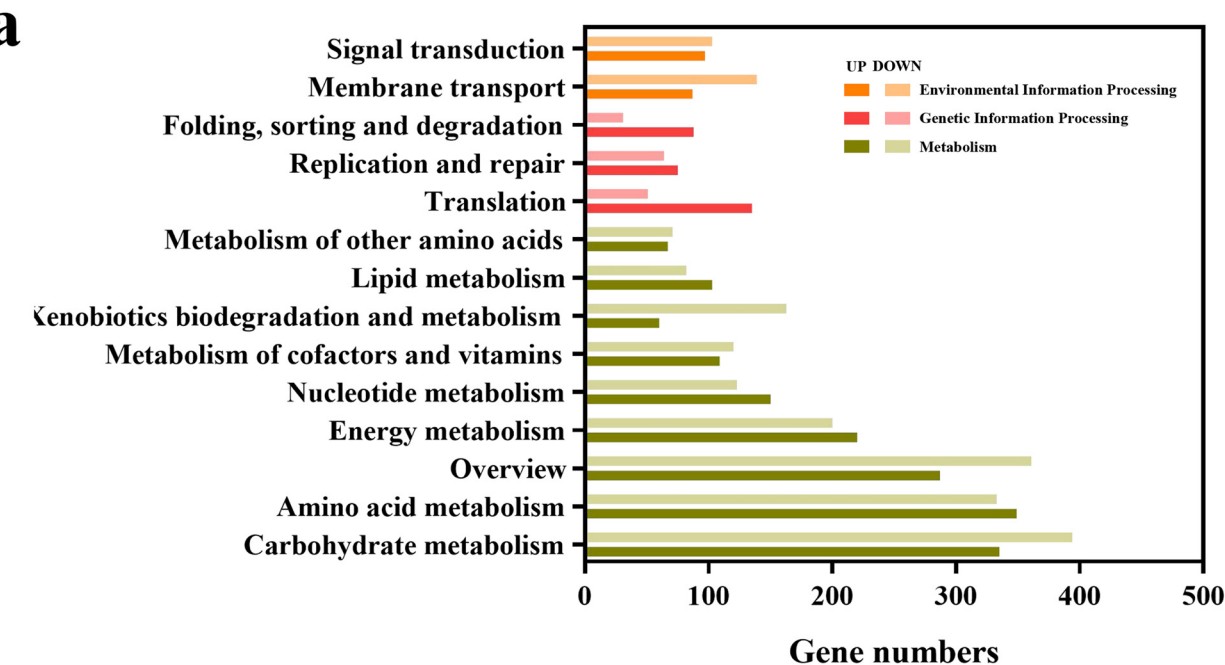

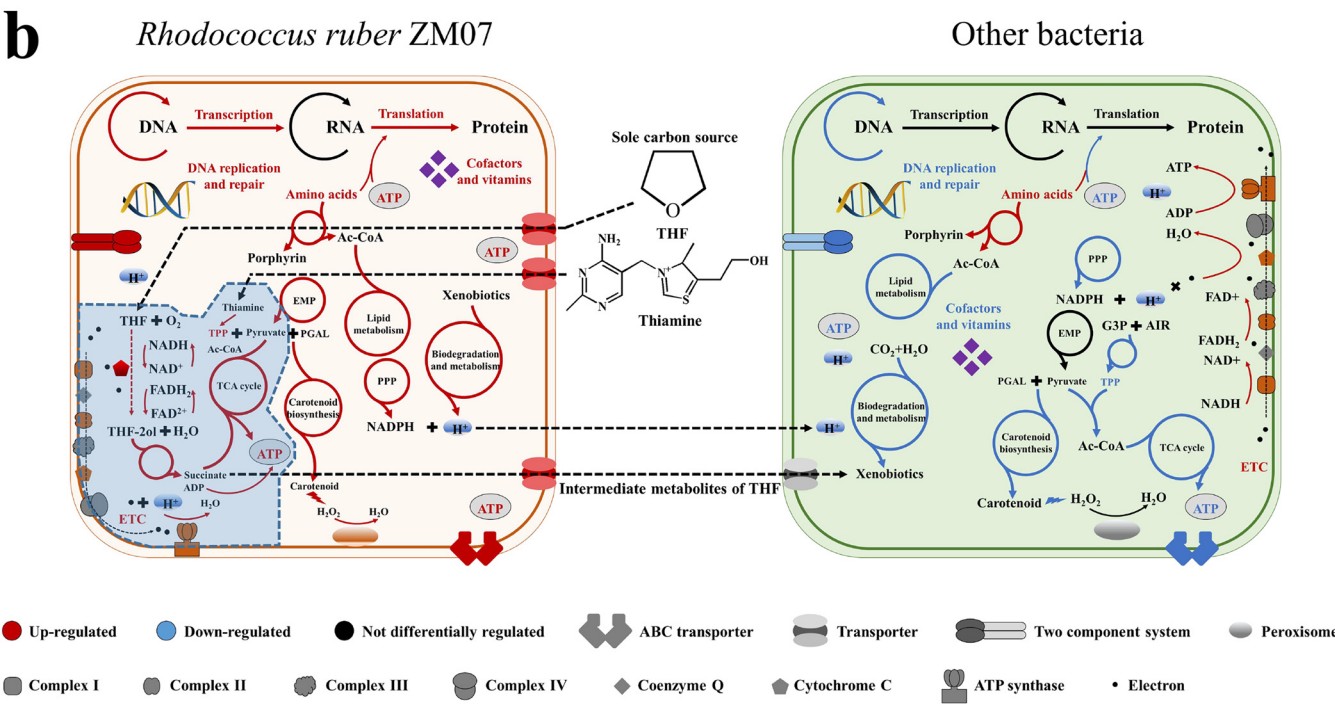

**FIG 4** (a) Results of KEGG classification analysis of differentially expressed genes between T40-THI and T40-CK. The dark bars indicate the numbers of significantly upregulated genes in microbial culture H-1 with exogenous thiamine, while the light bars indicate the numbers of significantly downregulated genes. (b) Proposed interactions between strain ZM07 and other bacteria with exogenous thiamine addition after 40 transfers based on metatranscriptomic analyses. Gene/transporter/metabolic pathways are shown as upregulated (red), downregulated (blue), or not differentially regulated (black) in the THI group relative to the CK group. The blue dashed line marks the THF degradation pathway and thiamine metabolic pathway of strain ZM07. The thiamine-auxotrophic THF-degrading bacterium ZM07 can grow rapidly with THF as the sole carbon source in the presence of exogenous thiamine. Meanwhile, strain ZM07 degrades THF and provides intermediate metabolites as carbon sources for other nondegrading bacteria. However, additional thiamine inhibits the metabolism of other bacteria in the microbial community.

degradation performance in various combinations of ZM07, ZM11, and ZM12 (Fig. 5b), indicating that strain ZM11 might be a good cooperator of ZM07.

We analyzed the responses of the thiamine-auxotrophic THF-degrading bacterium ZM07 and the non-THF-degrading bacterium ZM11 to exogenous thiamine through metatranscriptome sequencing data combined with the genome sequence data of strains ZM07 and

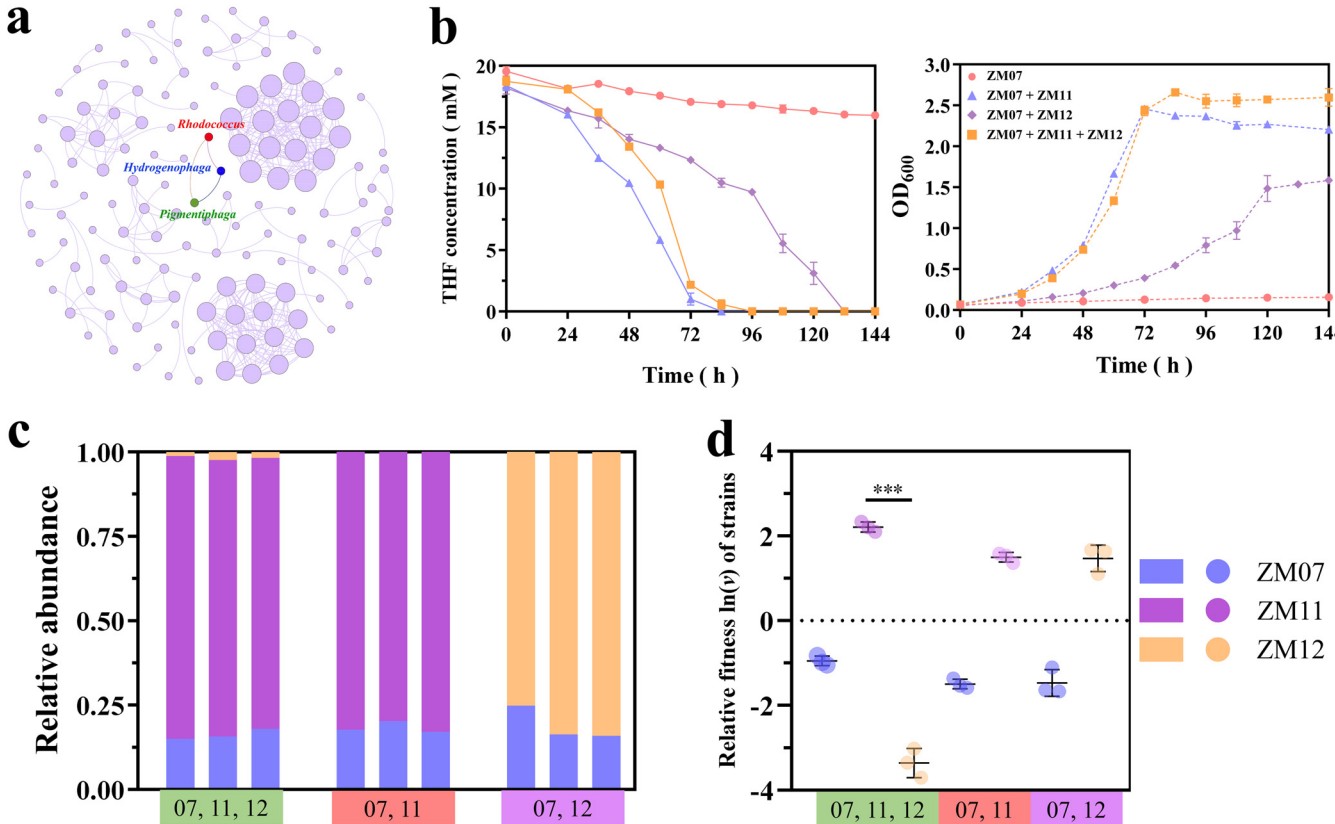

**FIG 5** (a) Network of co-occurring genera based on correlation analysis. A connection stands for a strong (Spearman's $\rho > 0.75$) and significant ($P < 0.05$) correlation. The size of each circle represents the degree value, which is the number of relationships between a genus and other genera in the network. (b) THF degradation and growth curves of monoculture (ZM07), two-strain systems (strains ZM07 and ZM11 and strains ZM07 and ZM12), and three-strain system (ZM07, ZM11, and ZM12) in the 3rd transfer. Relative abundances (c) and relative fitness [ln($v$)] (d) of ZM07, ZM11, and ZM12 in the two-strain systems (strains ZM07 and ZM11 and strains ZM07 and ZM12) and three-strain system (ZM07, ZM11, and ZM12) at 48 h in the 3rd transfer. The $P$ value indicates statistical significance between ZM07, ZM11, and ZM12 determined using Student's $t$ test ($n = 3$). ***, $P < 0.001$. Error bars represent standard errors of the means.

ZM11 sequenced earlier. The interactions between ZM07 and ZM11 changed with exogenous thiamine. According to the relative abundance, strain ZM11 grew better than ZM07 in the CK group but inversely with the presence of exogenous thiamine (Fig. 6b). Similarly, most DEGs of ZM07 were upregulated after addition of exogenous thiamine, but ZM11 showed the opposite trend (Fig. 6d). In total, 58 genes were upregulated in ZM07 but downregulated in ZM11 in the T1-THI group compared with the T1-CK group (Table S5), and similar relationships also existed in other groups (Fig. S9). Based on metatranscriptomic analyses, strains ZM07 and ZM11 also exhibited a seesaw phenomenon in metabolism (Fig. S10). Exogenous thiamine induced ZM07 to upregulate the genes involved in carbohydrate metabolism, amino acid metabolism, energy metabolism, lipid metabolism, cofactor and vitamin metabolism, replication, transcription, and translation but downregulated those in ZM11 (Fig. 6; Fig. S10; Table S5). In addition, the genes involved in propanoate metabolism were upregulated in ZM11, which would inhibit the metabolism of the bacteria and impede bacterial development (29). Overall, ZM07 was provoked by adding exogenous thiamine, but ZM11 was suppressed.

## DISCUSSION

Thiamine is an important compound in all living organisms. Its activated form, TPP, has a variety of auxiliary functions in cell metabolism. In addition, thiamine is widely used in agriculture (30) and livestock production (31). However, there have been few studies on adding exogenous thiamine to polluted environments to improve the efficiency of microbial remediation. In environmental bioremediation, pollutant-degrading

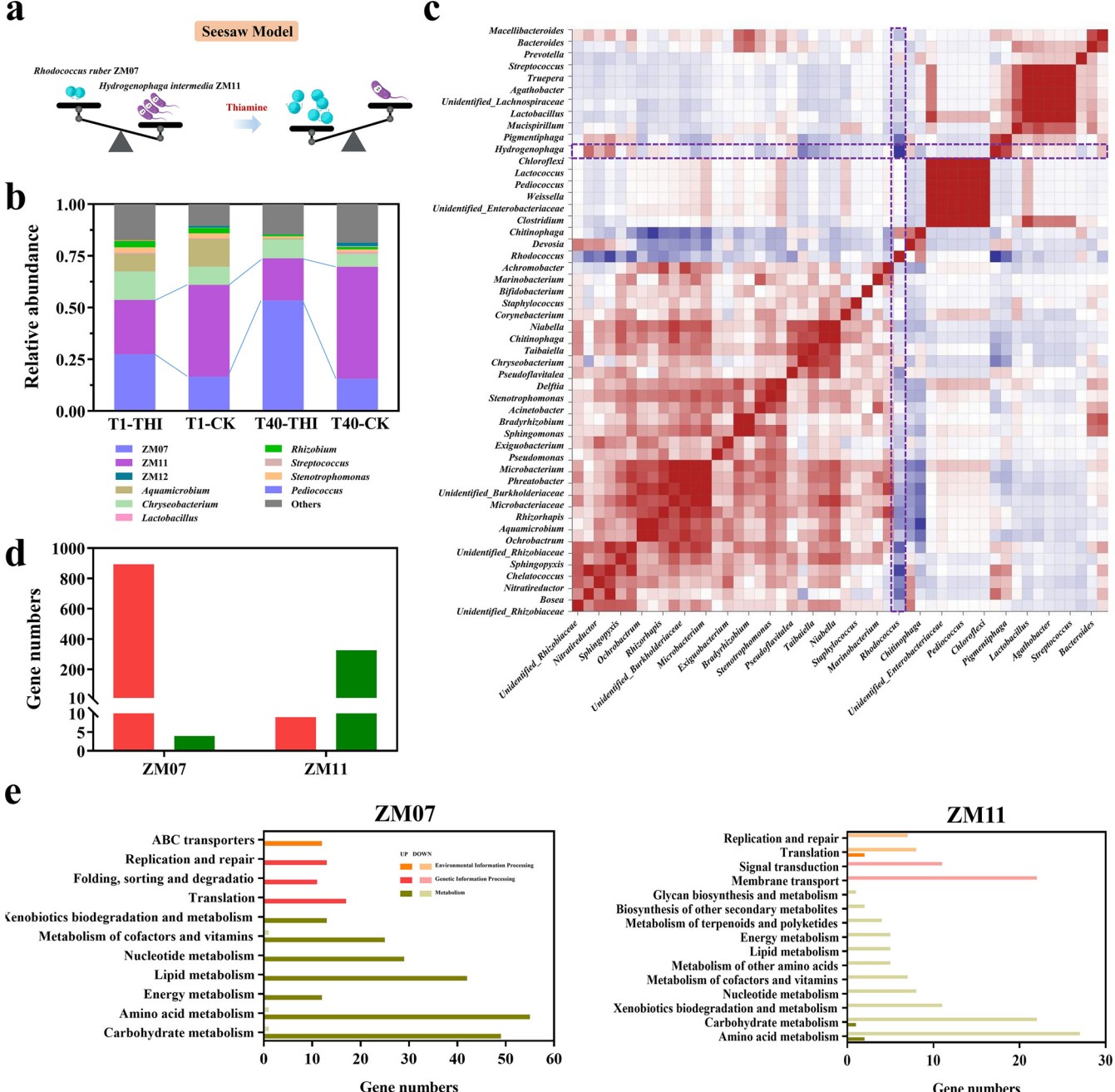

**FIG 6** (a) Schematic illustration of the seesaw model between ZM07 and ZM11. (b) Relative abundances of dominant microbial genera for different groups based on high-throughput sequencing. Less abundant phyla are grouped in the category "others." (c) Correlation heat map of different species. Blue indicates a negative correlation, and red indicates a positive correlation. The darker the color, the stronger the correlation between species. The dashed line emphasizes *Rhodococcus* and *Hydrogenophaga*. (d) Numbers of DEGs of ZM07 and ZM11 (T1-THI versus T1-CK). Red bars represent upregulated genes, and green bars represent downregulated genes. (e) Results of the KEGG classification analysis of DEGs from ZM07 (left) and ZM11 (right) (T1-THI versus T1-CK). Dark bars indicate the number of significantly upregulated genes in strain ZM07 or ZM11 with exogenous thiamine, while light bars indicate the number of significantly downregulated genes.

microbial communities involve cross-feeding patterns on growth factors such as vitamins and amino acids (32, 33). Interspecies interactions are essential for the composition and function of the microbiome (34). In our previous studies, we examined a THF-degrading microbial culture containing a thiamine-auxotrophic THF-degrading bacterium (24, 25). The results showed that exogenous thiamine can significantly improve the THF-degrading efficiency and biomass of H-1 (Fig. 1), indicating that thiamine could be used as a biological stimulus to promote bioremediation. This work

provides a new strategy for enhancing efficiency in the practical application of sewage treatment.

**Thiamine-mediated THF degradation.** The biological degradation efficiency of pure/mixed cultures is often affected by environmental conditions. Hence, understanding these mechanisms can be helpful for controlling biofunctional bacteria/communities and enhance their capability for bioremediation. *thm* is a known THF degradation gene cluster in ZM07 (35). As reported previously, Thm can catalyze THF to 2-hydroxytetrahydrofuran, and this hydroxylating step relies on NADH and oxygen (28). In this study, we found that the genes involved in NADH synthesis were significantly upregulated with exogenous thiamine in ZM07 (Table S4), and NADH is produced during an enzymatic reaction with TPP as a cofactor. We suspected that exogenous thiamine may improve the THF degradation efficiency of ZM07 by upregulating genes involved in NADH-producing pathways. Previous research has shown that both cytochrome *c* and oxygen act as surrogate electron acceptors of ThmD (the oxidoreductase of Thm), but the former has a higher priority (28). In this study, cytochrome *c* synthesis gene expression dramatically increased in ZM07 (Table S4), indicating the importance of this pathway in the degradation of THF. Accordingly, we predict that cytochrome *c* might be the major electron receptor in the initial step of THF hydroxylation, as previously reported (28). Therefore, the enhancement of the gene expression of the thiamine metabolic pathway could accelerate the THF degradation process, which shows important application potential to improve THF degradation through changes in important environmental factors in bioremediation.

**Gene expression associated with thiamine metabolism.** Thiamine, the coenzyme of $\alpha$-ketoglutarate dehydrogenase and pyruvate dehydrogenase, plays a strong role in carbohydrate metabolism and is an indispensable growth factor for maintaining normal cellular functions (36). Our previous study indicated that ZM07 is a thiamine-auxotrophic THF-degrading bacterium (25), which is the dominant strain performing the THF degradation function in microbial community H-1. Therefore, the growth and THF degradation performance of ZM07 relies on thiamine prototrophic microbes in the community. In this study, the changes in gene expression associated with thiamine metabolism of ZM07 and its prototrophic cooperators were observed in order to understand the response model of the functional microbial community to important growth factors. Pan et al. (13) proved that the thiamine concentration in the rumens of dairy cows was positively correlated with thiamine-relevant bacteria. Similar results were found in a study of the microbial community in the North Atlantic Ocean, and thiamine-related compounds were positively correlated with bacterial production (37). However, a recent study showed that thiamine metabolism genes in diatoms cannot be regulated by exogenous thiamine (38). Herein, the thiamine metabolic genes were significantly downregulated in some genera with exogenous thiamine (Fig. 3), and these strains might be the core members that produced and secreted thiamine in H-1 before thiamine was added. Conversely, exogenous thiamine made the cooperators of ZM07 redundant, intensively activating the thiamine metabolism process of ZM07 (Fig. 3) and successively making the THF degrader ZM07 dominate the microbial community. Growth factors have different effects on the related metabolic pathways of auxotrophic and prototrophic microbes, which offer the possibility of altering the interaction relationship of the microorganisms in functional microbial communities.

**Exogenous thiamine changed the microbial interaction pattern.** Microbial communities are the foundation of diverse ecosystems on Earth. However, the maintenance of such important communities may sometimes rely on a few or even a single growth factor, and the influence of the growth factors on the microbial community may vary greatly by habitat. Studies on rumen microbial communities have reported that thiamine supplementation can relieve subacute ruminal acidosis by increasing the rumen pH and regulating the rumen microbial community structure (39, 40). Additionally, exogenous thiamine can impact the competition of intestinal flora by altering the internal thiamine pool of gut microbes (41). However, the impacts of thiamine on environmental microbial communities are poorly explored, limiting our comprehensive understanding of thiamine functions. Herein, exogenous thiamine significantly decreased the pH of H-1 and had a great impact on the microbial community structure (Fig. 1 and 2; Fig. S1). The results suggested that H-1

may have responses to thiamine different from those of rumen or intestinal microbial communities.

Afterward, we investigated the effects of thiamine on microorganisms at the genetic level using metatranscriptome sequencing and analysis. Our results showed that exogenous thiamine changed the interaction pattern between auxotrophic THF-degrading bacterium ZM07 and non-THF-degrading bacteria. Under thiamine-poor conditions, the thiamine-auxotrophic THF-degrading bacterium ZM07 needs other members in this microbial community to provide thiamine, meanwhile, strain ZM07 can degrade THF (the sole carbon source) and supply intermediate metabolites as carbon sources for other nondegrading bacteria, as previously reported (25). However, the genes involved in most gene/transporter/metabolic pathways, such as THF degradation, tricarboxylic acid cycle, and DNA replication and repair, were upregulated in strain ZM07 with exogenous thiamine (Fig. 4b). Therefore, we speculated that strain ZM07 abandoned its cooperators and occupied most of the niches of H-1 members under thiamine-rich conditions. However, the genes involved in the biosynthesis of amino acids and ETC were upregulated in other bacteria, while most of the genes in basal metabolism were significantly downregulated (Fig. 4b), illustrating the weakening of other life activities (42). Therefore, we predicted that exogenous thiamine caused ZM07 to occupy most of the resources and niches in the community and that other bacteria must regulate the activity of crucial biological metabolic pathways to maintain their growth and existence. Overall, exogenous thiamine can redistribute resources by changing the microbial interaction pattern between THF-degrading bacteria and their cooperators, thereby affecting the THF degradation function of microbial community H-1.

**Two dominant strains, ZM07 and ZM11, exhibit a seesaw relationship.** The large fraction of auxotrophs in the microbiota implies their strong dependence on the exogenous supply of these micronutrients from prototrophic species (14, 43, 44). Based on topology analysis of networks, crucial species that have vital functions in the microbial community can be identified (45). Here, we studied the microbial interaction network of microbial community H-1 using network analysis, and strain ZM11 is one of the cooperators of ZM07 predicted by it. For the two principal microorganisms in H-1, ZM07 and ZM11, exogenous thiamine not only increased or decreased the relative abundance of (Fig. 6b) but also caused up- or downregulation of the genes involved in basal metabolism (Fig. 6e), indicating that thiamine shifted the pattern of interaction between microorganisms and reshaped the microbial community. Therefore, we propose a seesaw relationship between ZM07 and ZM11 in this study (Fig. 6a). Notably, the genes related to propanoate metabolism were significantly upregulated in ZM11, while a large number of metabolic pathways were inhibited. According to a previous study, propionate strongly inhibits cell growth and can be degraded through the 2-methylcitric acid cycle in both prokaryotes and eukaryotes (29, 46). Therefore, we considered that energy metabolism in ZM11 was inhibited by exogenous thiamine, leading to the accumulation of propionate and causing side effects on bacterial growth. In microbial communities, mutualism could promote communication between bacterial species to obtain an environmental niche and nutrients (47); however, the interaction between microorganisms could be changed by environmental factors (48). Here, exogenous thiamine changed the microbial co-occurrence pattern between strains ZM07 and ZM11. The thiamine-auxotrophic THF-degrading bacterium ZM07 could compete for resources and niches and suppress cooperator survival with abundant thiamine in the environment, leading to modifications in the structure and function of the microbial community.

In conclusion, thiamine could affect the interaction patterns between the thiamine-auxotrophic THF-degrading bacterium ZM07 and other microorganisms by altering resource and niche distributions, thereby improving the THF degradation efficiency of H-1. Our findings show that adding special micronutrients might be an effective way to increase the biodegradation efficiency of THF or other pollutants. Additionally, two dominant strains, ZM07 and ZM11, exhibit a seesaw pattern, with completely different responses to exogenous thiamine. The findings of this study not only provide new

insight into reveal the effect of thiamine on the environmental microbiota but also provide evidence for employing micronutrients as a potential method to regulate and improve the biodegradation of pollutants in microbial communities.

## MATERIALS AND METHODS

**Microbial culture experiments.** The THF-degrading microbial culture H-1 was obtained in our earlier work (24). Two milliliters of H-1 was transferred into 100 mL of sterile basal salt medium (BSM; see the supplemental material) (49) with 20 mM THF. H-1 was cultivated in BSM for 3 days, collected by centrifugation (7,000 × $g$, 10 min), washed three times, and resuspended in sterile BSM ($OD_{600}$ = 6); 1-mL cell suspensions were added to 100 mL BSM with 20 mM THF as the 1st transfer. Here, two treatment groups were established: the group with no exogenous thiamine was the control (CK) group (T1-CK and T40-CK, which represent samples of the 1st and 40th transfers, respectively), and the group with 0.01 mM added thiamine was the experimental (THI) group (T1-THI and T40-THI, which represent samples of the 1st and 40th transfers, respectively). These two groups were transferred in the same way 40 times. Bacteria were cultured at 30°C and 160 rpm, and samples were collected at 72 h cultivation every 5 transfers for further analysis. The chemicals used in this study are listed in the supplemental material.

**DNA extraction and 16S rRNA gene sequencing.** Samples of microbial community H-1 were collected at 72 h in the 1st and 40th transfers with or without exogenous thiamine (see Table S1 for details). Genomic DNA of the samples was extracted by using the E.Z.N.A. bacterial DNA kit (Omega Biotek, USA), and its quantity and quality were measured by using a NanoDrop 2000c spectrophotometer (Thermo Scientific, USA). The bacterial 16S rRNA gene (V3-V4 region) was amplified by using Phusion high-fidelity PCR master mix (New England Biolabs). The microbiota composition of the samples was sequenced by Novogene Bioinformatics Technology Co., Ltd. (Beijing, China), on an Ion S5 XL platform (Thermo Fisher).

**Network construction.** The co-occurrence network was constructed using the method described by Wang et al. (50) based on the Spearman correlation matrix created with the vegan (51), igraph (52), and Hmisc (53) packages in the R environment. A correlation between two items was considered statistically robust if the Spearman's correlation coefficient was >0.75 and the $P$ value was <0.05. The nodes in this network represent genera, and the edges that connect these nodes represent co-occurrence between genera. Network visualization was conducted on the interactive platform of Gephi (54).

**Network-directed isolation of the cooperators of strain ZM07 and coculture experiments.** In our previous work, *Rhodococcus ruber* strain ZM07 (THF-degrading bacterium, collection number CCTCC AB 2019217) was isolated from an enrichment culture H-1 (25). The 16S rRNA gene sequence of ZM07 was compared with the gene library to locate the node corresponding to it in the network. Then, we identified two nodes connected to strain ZM07. According to the results of network analysis, two non-THF-degrading strains, *Hydrogenophaga intermedia* ZM11 (collection number CCTCC M 2021639) and *Pigmentiphaga daeguensis* ZM12 (collection number CCTCC M 2021640), were isolated using the gradient dilution technique on selective culture medium, i.e., the cell suspension of microbial culture H-1 was diluted to $10^{-1}$, $10^{-2}$, $10^{-3}$, $10^{-4}$, $10^{-5}$, $10^{-6}$, and $10^{-7}$, and microbes were then cultured and isolated using the surface spread plate method. Pantothenate medium (55) and R2A agar (56, 57) were used to isolate strains ZM11 and ZM12, respectively. For cultivation medium ingredients, see the supplemental material. After isolation, 16S rRNA genes were amplified by PCR using the universal bacterial primers 27F and 1492R. The 16S rRNA sequence obtained was aligned to sequences in GenBank using the BLAST program.

Strains ZM07, ZM11, and ZM12 were initially incubated in lysogeny broth (LB) culture medium. Cells were collected by centrifugation (7,000 × $g$, 10 min), washed three times and resuspended in sterile BSM ($OD_{600}$ = 3); 100 mL of BSM with 20 mM THF was inoculated with 1 mL of ZM07 and 1 mL of ZM11 or ZM12 for two-strain coculture, and 1 mL of ZM07 and 0.5 mL of each ZM11 and ZM12 for three-strain coculture as the 1st transfer. Subsequently, 1 mL of the cocultured systems ($OD_{600}$ = 6) was transferred to fresh medium in the same way for three times. All the bacteria were cultured at 30°C and 160 rpm. In the 3rd transfer, samples were collected for determination of the residual THF concentrations as well as the biomass ($OD_{600}$) every 24 h; in addition, cocultured samples were collected for analysis of the relative strain abundances at 48 h and 72 h, respectively.

**RNA extraction, sequencing, sequence analysis, and annotation.** RNA of samples was extracted by using the RNeasy minikit (Qiagen, Germany). Then, RNA was purified and concentrated using an RNase-Free DNase set and an RNeasy MinElute cleanup kit (Qiagen, Germany), respectively. Subsequently, the concentrated RNA was converted to double-stranded cDNA using the FastKing RT kit (Tiangen, Beijing, China) according to the manufacturer's instructions. Metatranscriptomic libraries were constructed and sequenced by Novogene Bioinformatics Technology Co., Ltd. (Beijing, China), on an Illumina NovaSeq 6000 platform (Illumina). Raw data were processed for *de novo* transcriptomic assemblies. The read count data were further used for gene differential expression analysis, and the significantly differentially expressed genes (DEGs) between groups (T1-THI versus T1-CK, T40-CK versus T1-CK, T40-THI versus T1-THI, and T40-THI versus T40-CK) were identified using $P$ values (58). Additionally, these genes were searched and annotated in the KEGG database (59). For additional details of cDNA library construction, transcriptome sequencing (RNA-seq), and data analysis, see the supplemental material.

**Detection of THF and composition of strains in coculture systems and statistical analysis.** THF concentration was determined according to our previously described method using a gas chromatograph equipped with a flame ionization detector (FID) and an AOC-20i autoinjector (GC-2014C; Shimadzu, Japan) (25). The degradation ratio was calculated according to our previously described formula (24). The biomass of the samples was monitored by recording the $OD_{600}$ using a UV-3100PC spectrometer (Mapada, China).

The relative abundances of ZM07, ZM11, and ZM12 were determined by using quantitative PCR (qPCR). Total genomic DNA of the bacterial culture was extracted using the method mentioned above. Primers for the qPCR assay are shown in Table S2, and the qPCR were performed as we previously reported (25). Using the $2^{-\Delta\Delta CT}$ method (60), the gene abundance level of different bacteria was calculated. Then, the relative fitness ($v$) of each strain was calculated as follows: $[a_1 \times (1 - a_0)]/[a_0 \times (1 - a_1)]$, where $a_0$ and $a_1$ represent the initial and final frequencies of each strain, respectively. To obtain normally distributed residuals, all fitness values were ln transformed. Values of ln($v$) that are >0 or <0 indicate that the strain frequency increased or decreased relative to its competitor in the same cocultured system, respectively (61, 62).

**Data availability.** 16S rRNA gene sequencing data were deposited in the China National GeneBank Sequence Archive (CNSA) of the China National GeneBank DataBase (CNGBdb) under accession number CNP0001620. Metatranscriptome sequencing data were deposited in the CNSA of CNGBdb with accession number CNP0001609. The genomes of ZM07, ZM11, and ZM12 were deposited in the CNSA of CNGBdb with accession numbers CNP0001612, CNP0001636, and CNP0001637.

## SUPPLEMENTAL MATERIAL

Supplemental material is available online only.

**SUPPLEMENTAL FILE 1**, PDF file, 3.4 MB.

## ACKNOWLEDGMENTS

This research was funded by grants from the National Natural Science Foundation of China, grants 32170107 and 41721001, and the Key Research and Development Program of Zhejiang Province, grant 2021C03168.

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
