## [Reviewer comments · Microbiology Spectrum]

Microbiology Spectrum

Thiamine-mediated microbial interaction between auxotrophic *Rhodococcus ruber* ZM07 and prototrophic cooperators in a tetrahydrofuran-degrading microbial community H-1

Hui Huang, hao WU, Minbo Qi, Haixia Wang, and Zhenmei Lu

Corresponding Author(s): Zhenmei Lu, Zhejiang University

Review Timeline:

Submission Date:	November 15, 2022
Editorial Decision:	February 18, 2023
Revision Received:	March 31, 2023
Accepted:	April 10, 2023

Editor: Victor Gonzalez

Reviewer(s): Disclosure of reviewer identity is with reference to reviewer comments included in decision letter(s). The following individuals involved in review of your submission have agreed to reveal their identity: Reid Simmer (Reviewer #1)

Transaction Report:

DOI: <https://doi.org/10.1128/spectrum.04541-22>

February 18, 2023

Prof. Zhenmei Lu
Zhejiang University
Institute of Microbiology, College of Life Science, Zhejiang University
Hangzhou, Zhejiang 310058
China

Re: Spectrum04541-22 (Thiamine-mediated microbial interaction between auxotrophic *Rhodococcus ruber* ZM07 and prototrophic cooperators in a tetrahydrofuran-degrading microbial community H-1)

Dear Prof. Zhenmei Lu:

Link Not Available

Sincerely,

Victor Gonzalez

Journals Department
Reviewer comments:

Reviewer #1 (Comments for the Author):

Review

The manuscript summarizes interesting results from a novel study on an important topic, which addressed a key knowledge gap that will help guide future research. It is timely as tetrahydrofuran is a common industrial solvent and environmental pollutant. This study expands on the authors' previous work with thiamine auxotroph and tetrahydrofuran metabolizer, *Rhodococcus ruber* ZM07. The current study examines shifts in microbial communities containing ZM07 through passage with and without exogenous thiamine. The authors then utilize a prediction network to identify two strains closely linked to ZM07, one of which, ZM11, supports ZM07 through thiamine biosynthesis. The authors' findings are well supported by genomic analyses and statistical tests.

Main Comments

1. The network model proved successful in identifying strains *Hydrogenophaga intermedia* ZM11 and *Pigmentiphaga daeguensis* ZM12. However, a detailed description of model construction and utilization are absent from the methods section and is only briefly mentioned in the supplementary materials. Detailed methods are needed to explain how the network was used to identify and isolate ZM11 and ZM12. As this is a significant finding in the paper, these details should be described in the main text.
2. The methods state that tetrahydrofuran was the only carbon source provided in liquid cultures. However, the results indicate growth in cultures both with and without exogenous thiamine, suggesting that the community provided ZM07 with thiamine. What carbon source(s) did these community members utilize? If it is assumed that THF metabolites supported these strains, it is not well explained in the text. Consider emphasizing this point. Furthermore, were you able to measure these metabolites in significant quantities? Or were you able to measure thiamine produced by community members in the bulk media? Otherwise, this phenomenon is only speculative and should be stated as such.
3. To emphasize the importance of this research, consider adding perspective how this work can be applied in remediation. How can the findings of this study be applied to field bioremediation of tetrahydrofuran and other xenobiotic compounds? Consider adding this in more detail in the 'Importance' section and at the end of the discussion beyond line 294.

Specific Comments

1. Line 13: Run-on sentence, better wording is needed.
2. Line 21: "the network" - What network? Need to introduce network analysis earlier in abstract.
3. Line 22: "was proved" - better working needed.
4. Lines 23-26: run-on sentence, consider rewording.
5. Line 26: "quite the opposite"
6. Line 47: "a thiazole ring and a pyrimidine ring."
7. Line 55: "others" - vague, consider revising.
8. Lines 70-71: Consider adding additional description of the specific problems THF pollution is causing.
9. Line 72: Consider introducing your previous work (enrichment of H-1) in-text
10. Line 86: microbial culture H-1 was transferred
11. Line 87: "different THF degradation ability with different growth curve" - needs rewording
12. Lines 92-95: Lengthy sentence, consider revising
13. Figure 2d: very difficult to distinguish columns between treatment groups (T1-CK, T1-THI, T40-CK, T40-THI). Consider adding spaces between groups. Same for Figure S2.
14. Figure 2e: The similarity squares along the bottom axis are very difficult to interpret. Furthermore, the short explanation in text "In addition, the community heatmap of samples showed that T40-CK is closer to T1-CK than T1-THI" offers little interpretation. Can this relationship be instead proven through statistical analysis? At the very least, the graphic needs to be revised to improve clarity and in-text explanation needs to be expanded. In addition, units should be added to the color spectrum legend.
15. Line 115: Unclear what "transfer" is referring to.
16. Line 119: Define DEG the first time it is used in each section.
17. Line 120: "A plenty" needs to be reworded.
18. Line 121: "carbon metabolism, ribosome, biosynthesis of amino acids," -awkward phrasing, needs to be reworded
19. Line 122: "ABC transporters, two-component system, purine metabolisms" -awkward phrasing, needs to be reworded

20. Lines 123-126: Awkward transition between paragraphs, consider removing the last sentence in the previous paragraph (Lines 123-125)
21. Line 126: "thiamine metabolic metabolism"
22. Lines 130-131: "was significantly up-regulated after adding exogenous thiamine in ZM07" -reference figure to support statement.
23. Figures S3a and 3a: Volcano plots are difficult to interpret. The difference between treatment and control groups is indistinguishable. Consider adding separate plots for THI and CK groups and explanation in the captions. Otherwise, consider removing altogether. Also, define DEGs in captions.
24. Figures S3b, 3b, and 5e: Neither the legend nor caption identifies which color corresponds to each treatment group (THI or CK). Captions in S3b and 3b need more detail. Also, define DEGs the first time it is used in each caption.
25. Line 140: "The results we have seen show significant differences" awkward phrasing, needs rewording.
26. Lines 143 and 146: "significantly regulated by exogenous thiamine" and "significantly up-regulated in ZM07 with exogenous thiamine" - significant by what criteria?
27. Line 147: "(one possible electron acceptor of Thm)" - needs a citation or wait to mention in discussion.
28. Figure 3c: Needs further explanation in caption and in text.
29. Lines 150-151: "the situation in other bacteria was opposite" - needs rewording.
30. Figure S7 - Needs further explanation in caption and in text. Legend needs units if appropriate.
31. Lines 203-204: "In our previous studies, we examined We obtained a THF-degrading microbial culture containing a thiamine auxotrophic THF-degrading bacterium in our previous studies"
32. Line 216: Cut "By the way"
33. Line 219: "cytochrome c synthesis gene expression had a dramatically increase in ZM07" - Also consider adding reference to a figure to support this.
34. Line 246: "the heart of all ecosystems, same growth factors" -needs rewording
35. Line 247: "Rumen microbes' " - needs rewording
36. Line 254: Consider starting a new paragraph before "Afterwards,"
37. Line 297: "our earlier period work"
38. Line 300: "1 mL cell suspensions were"
39. Line 301: What chemical form of thiamine was added? Thiamine HCl or thiamine nitrate?
40. Line 301: "one group was added with 0.01 mM thiamine"
41. Line 303: "and the other one was not" - needs rewording.
42. Line 320-321: "using conventional method." Citation needed, (e.g., per manufacturer's instructions).
43. Line 331- Description or citation needed for the "gradient dilution technique."
44. Line 334- "16S rRNA genes were amplified by PCR, using the universal bacterial primers 27F_and 1492R."
45. Line 344: "1 mL of the cocultured systems"
46. Line 351: What detector was used with the GC to measure THF?
47. Line 353: What wavelength was used to measure optical density?

Reviewer #2 (Comments for the Author):

The manuscript "Thiamine-mediated microbial interaction between auxotrophic *Rhodococcus ruber* ZM07 and prototrophic cooperators in a tetrahydrofuran-degrading microbial community H-1" (control no. Spectrum04541-22) by Hui Huang and colleagues is experimentally sound and provides interesting data.

However, there are several major drawbacks that must be addressed ahead of my agreement on publication in *Microbiology Spectrum*.

First, there are some discrepancies between results and their interpretation. For example, the authors state in the abstract that "Our results indicate.....by participating in NADH producing pathway...". Although there is some evidence for this hypothesis from the metatranscriptome data, this conclusion should be weakened. Then, the wording "...participating in NADH producing pathway..." is misleading. Is thiamine a cofactor here, or do the authors mean that thiamine upregulates genes involved in NADH-producing pathways? Please clarify in the abstract by rewriting.

Second, the results sections on metatranscriptome analysis with respect to conclusions derived from metabolic data needs to be revised regarding its interpretation. A good beginning is made in lines 193-195. Explain why which metabolic changes are hypothesized to reduce or improve the growth of individual strains.

Third, thorough check of English writing and spelling is required

Fourth, the manuscript is not well written and is therefore hard to read.

Examples only, but the whole manuscript must be carefully checked and rewritten:

1. The supplementary figures (at least in the PDF) need to be enlarged.
2. The figures should be numbered for clarity.
3. Lines 36-37: better "Thiamine may help ZM07 to occupy..."
4. Line 214-216: Cited Figure 3c does not show the genes upregulated by thiamine. The genes involved in NADH synthesis, which were found up-regulated, should be mentioned here or elsewhere.
5. Lines 87, 98, 99: all abbreviations such as THI, CK, T40 etc. must be introduced.
6. Line 90: do not start sentences with "And"
7. Line 93: what is "more excellent"? Quantify by fold change.
8. Compared TO.
9. Line 99: "...during transferring". Please rewrite.
10. Line 102: -diversity.
11. Line 118: ...many significant alterations in the microbial community: too general statement; composition or metatranscriptome? Which alterations in detail? Start with the data and end with such a (more precisely reworded) conclusion
12. Line 119: Define DEG (differentially expressed genes)
13. Line 137: Consider Figure S5 as a full figure due to its importance.
14. Line 147, 153 and elsewhere: "facilitate" is not appropriate here, better "result in upregulation of genes involved in..."
15. Line 156: ...isolation of bacteria related to strain...
16. Line 161: Fortunately? Please describe how the two target bacteria were isolated.
17. Lines 162-163: There are much more species and strains in these genera. Do you mean "...in the microbiota analysed here?"
18. Line 165 Strain ZM07 exhibits...; "much better" is not an appropriate qualification here, please quantify.
19. Line 175, 266: Here, we propose a seesaw model...
20. Line 215, Fig. 3c does not indicate the genes mentioned in the text.
21. Fig. 1: Red and black colours indicate....
22. Fig. 2: please explain the scale of the heatmap (1e-1 etc.?)
23. ...and many further examples.

Staff Comments:

Preparing Revision Guidelines

- Point-by-point responses to the issues raised by the reviewers in a file named "Response to Reviewers," NOT IN YOUR COVER LETTER.
- Upload a compare copy of the manuscript (without figures) as a "Marked-Up Manuscript" file.

- Each figure must be uploaded as a separate file, and any multipanel figures must be assembled into one file.
- Manuscript: A .DOC version of the revised manuscript
- Figures: Editable, high-resolution, individual figure files are required at revision, TIFF or EPS files are preferred

Please return the manuscript within 60 days; if you cannot complete the modification within this time period, please contact me. If you do not wish to modify the manuscript and prefer to submit it to another journal, please notify me of your decision immediately so that the manuscript may be formally withdrawn from consideration by Microbiology Spectrum.

Thiamine-mediated microbial interaction between auxotrophic *Rhodococcus ruber* ZM07 and prototrophic cooperators in a tetrahydrofuran-degrading microbial community H-1

Hui Huang^{1,2}, Hao Wu¹, Minbo Qi¹, Haixia Wang¹, Zhenmei Lu^{1*}

¹MOE Laboratory of Biosystem Homeostasis and Protection, College of Life Sciences, Zhejiang University, Hangzhou, 310058, China

²Institute of Translational Medicine, Zhejiang University, Hangzhou, 310029, China

Review

The manuscript summarizes interesting results from a novel study on an important topic, which addressed a key knowledge gap that will help guide future research. It is timely as tetrahydrofuran is a common industrial solvent and environmental pollutant. This study expands on the authors' previous work with thiamine auxotroph and tetrahydrofuran metabolizer, *Rhodococcus ruber* ZM07. The current study examines shifts in microbial communities containing ZM07 through passage with and without exogenous thiamine. The authors then utilize a prediction network to identify two strains closely linked to ZM07, one of which, ZM11, supports ZM07 through thiamine biosynthesis. The authors' findings are well supported by genomic analyses and statistical tests.

Main Comments

1. The network model proved successful in identifying strains *Hydrogenophaga intermedia* ZM11 and *Pigmentiphaga daeguensis* ZM12. However, a detailed description of model construction and utilization are absent from the methods section and is only briefly mentioned in the supplementary materials. Detailed methods are needed to explain how the network was used to identify and isolate ZM11 and ZM12. As this is a significant finding in the paper, these details should be described in the main text.
2. The methods state that tetrahydrofuran was the only carbon source provided in liquid cultures. However, the results indicate growth in cultures both with and without exogenous thiamine, suggesting that the community provided ZM07 with thiamine. What carbon source(s) did these community members utilize? If it is assumed that THF metabolites supported these strains, it is not well explained in the text. Consider emphasizing this point. Furthermore, were you able to measure these metabolites in significant quantities? Or were you able to measure thiamine produced by community members in the bulk media? Otherwise, this phenomenon is only speculative and should be stated as such.
3. To emphasize the importance of this research, consider adding perspective how this work can be applied in remediation. How can the findings of this study be applied to field bioremediation of tetrahydrofuran and other xenobiotic compounds? Consider adding this in more detail in the 'Importance' section and at the end of the discussion beyond line 294.

Specific Comments

1. Line 13: Run-on sentence, better wording is needed.

2. Line 21: “the network” – What network? Need to introduce network analysis earlier in abstract.
3. Line 22: “was proved” – better working needed.
4. Lines 23-26: run-on sentence, consider rewording.
5. Line 26: “quite **the** opposite”
6. Line 47: “**a** thiazole ring and **a** pyrimidine ring.
7. Line 55: “others” – vague, consider revising.
8. Lines 70-71: Consider adding additional description of the specific problems THF pollution is causing.
9. Line 72: Consider introducing your previous work (enrichment of H-1) in-text
10. Line 86: microbial culture H-1 **was** transferred
11. Line 87: “different THF degradation ability with different growth curve” – needs rewording
12. Lines 92-95: Lengthy sentence, consider revising
13. Figure 2d: very difficult to distinguish columns between treatment groups (T1-CK, T1-THI, T40-CK, T40-THI). Consider adding spaces between groups. Same for Figure S2.
14. Figure 2e: The similarity squares along the bottom axis are very difficult to interpret. Furthermore, the short explanation in text “In addition, the community heatmap of samples showed that T40-CK is closer to T1-CK than T1-THI” offers little interpretation. Can this relationship be instead proven through statistical analysis? At the very least, the graphic needs to be revised to improve clarity and in-text explanation needs to be expanded. In addition, units should be added to the color spectrum legend.
15. Line 115: Unclear what “transfer” is referring to.
16. Line 119: Define DEG the first time it is used in each section.
17. Line 120: “A plenty” needs to be reworded.
18. Line 121: “carbon metabolism, ribosome, biosynthesis of amino acids,” -awkward phrasing, needs to be reworded
19. Line 122: “ABC transporters, two-component system, purine metabolisms” -awkward phrasing, needs to be reworded
20. Lines 123-126: Awkward transition between paragraphs, consider removing the last sentence in the previous paragraph (Lines 123-125)
21. Line 126: “thiamine **metabolic metabolism**”

22. Lines 130-131: “was significantly up-regulated after adding exogenous thiamine in ZM07” -reference figure to support statement.
23. Figures S3a and 3a: Volcano plots are difficult to interpret. The difference between treatment and control groups is indistinguishable. Consider adding separate plots for TH1 and CK groups and explanation in the captions. Otherwise, consider removing altogether. Also, define DEGs in captions.
24. Figures S3b, 3b, and 5e: Neither the legend nor caption identifies which color corresponds to each treatment group (TH1 or CK). Captions in S3b and 3b need more detail. Also, define DEGs the first time it is used in each caption.
25. Line 140: “The results we have seen show significant differences” awkward phrasing, needs rewording.
26. Lines 143 and 146: “significantly regulated by exogenous thiamine” and “significantly up-regulated in ZM07 with exogenous thiamine” – significant by what criteria?
27. Line 147: “(one possible electron acceptor of Thm)” – needs a citation or wait to mention in discussion.
28. Figure 3c: Needs further explanation in caption and in text.
29. Lines 150-151: “the situation in other bacteria was opposite” - needs rewording.
30. Figure S7 - Needs further explanation in caption and in text. Legend needs units if appropriate.
31. Lines 203-204: “~~In our previous studies, we examined~~ ~~We obtained~~ a THF-degrading microbial culture containing a thiamine auxotrophic THF-degrading bacterium ~~in our previous studies~~”
32. Line 216: Cut “By the way”
33. Line 219: “cytochrome c synthesis gene expression had a dramatically increase in ZM07” – Also consider adding reference to a figure to support this.
34. Line 246: “the heart of all ecosystems, same growth factors” -needs rewording
35. Line 247: “Rumen microbes’ “ – needs rewording
36. Line 254: Consider starting a new paragraph before “Afterwards,”
37. Line 297: “our earlier ~~period~~ work”
38. Line 300: “1 mL cell suspensions ~~were~~”
39. Line 301: What chemical form of thiamine was added? Thiamine HCl or thiamine nitrate?
40. Line 301: “one group ~~was added~~-with 0.01 mM thiamine”

41. Line 303: “and the other one was not” – needs rewording.
42. Line 320-321: “using conventional method.” Citation needed, (e.g., per manufacturer’s instructions).
43. Line 331- Description or citation needed for the “gradient dilution technique.”
44. Line 334- “16S rRNA genes **were** amplified by PCR₇ using the universal bacterial primers **27F_and** 1492R.”
45. Line 344: “1 mL **of** the cocultured systems”
46. Line 351: What detector was used with the GC to measure THF?
47. Line 353: What wavelength was used to measure optical density?
48. Supplementary Materials, Line 1: “**SUPPLEMENTARY MEYTHODS**”

Spectrum04541-22 (Thiamine-mediated microbial interaction between auxotrophic *Rhodococcus ruber* ZM07 and prototrophic cooperators in a tetrahydrofuran-degrading microbial community H-1)

Response to editor:

Thank you very much for your kindness in reviewing our manuscript. We also appreciate the insightful comments and suggestions of the two anonymous reviewers, which are very valuable for us to improve this manuscript. We have carefully considered each point brought up by two reviewers, and tried to answer their questions, and/or explain our thoughts. We believe that the quality and readability of this manuscript have now been significantly improved.

We have revised the manuscript to address all the comments provided.

By considering the reviewers' comments, the following major changes have been made in the text.

- (1) The English writing of this manuscript was improved to ensure clarity and coherence, all the spelling, grammar, and word choice errors in the manuscript were corrected.
- (2) The **ABSTRACT** and **IMPORTANCE** sections were rewritten according to the reviewer's comment.
- (3) The method about cooccurrence network constructed and the **Fig. S5** (**Fig. 3** in new manuscript) were moved to the text.
- (4) Some inappropriate descriptions were modified especially in the **RESULTS** section.

Reviewer comments:

Reviewer #1:

The manuscript summarizes interesting results from a novel study on an important topic, which addressed a key knowledge gap that will help guide future research. It is timely as tetrahydrofuran is a common industrial solvent and environmental pollutant. This study expands on the authors' previous work with thiamine auxotroph and tetrahydrofuran metabolizer, *Rhodococcus ruber* ZM07. The current study examines shifts in microbial communities containing ZM07 through passage with and without exogenous thiamine. The authors then utilize a prediction network to identify two strains closely linked to ZM07, one of which, ZM11, supports ZM07 through thiamine biosynthesis. The authors' findings are well supported by genomic analyses and statistical tests.

Main Comments

1. The network model proved successful in identifying strains *Hydrogenophaga intermedia* ZM11 and *Pigmentiphaga daeguensis* ZM12. However, a detailed description of model construction and utilization are absent from the methods section and is only briefly mentioned in the supplementary materials. Detailed methods are needed to explain how the network was used to identify and isolate ZM11 and ZM12. As this is a significant finding in the paper, these details should be described in the main text.

Response: Thank you for your suggestion. We have added detailed methods to explain how the network was used to identify and isolate ZM11 and ZM12. Firstly, the cooccurrence network was constructed based on the Spearman correlation matrix created with the VEGAN (1), igraph (2) and Hmisc (3) packages in the R environment. A correlation between two items was considered statistically robust if

the Spearman's correlation coefficient was > 0.75 and the p value was < 0.05 . The nodes in this network represent genera, and the edges that connect these nodes represent cooccurrence between genera. Network visualization was conducted on the interactive platform of Gephi (4). Then, the 16S rRNA gene sequence of strain ZM07 was compared with gene library to locate the node corresponding to it. Because strain ZM07 has been isolated and its genome has been sequenced in our previous work, it is easy to obtain its 16S rRNA gene sequence. The sequence data of ZM07 are now available in the China National GeneBank Sequence Archive (CNSA) of the China National GeneBank DataBase (CNGDB) with project codes CNP0001612. Finally, we identified two nodes connected to the strain ZM07. According to the 16S rRNA gene sequence corresponding to the nodes, these two non-THF-degrading strains ZM11 and ZM12 were isolated using selective medium. We also adjusted the order of the **“Network-directed isolation of the cooperators of strain ZM07 and coculture experiments”** and **“RNA extraction, sequencing, sequence analysis and annotation”** to make it more coherent. Please check this modification in lines 356-391 of the new manuscript, thank you.

2. The methods state that tetrahydrofuran was the only carbon source provided in liquid cultures. However, the results indicate growth in cultures both with and without exogenous thiamine, suggesting that the community provided ZM07 with thiamine. What carbon source(s) did these community members utilize? If it is assumed that THF metabolites supported these strains, it is not well explained in the text. Consider emphasizing this point. Furthermore, were you able to measure these metabolites in significant quantities? Or were you able to measure thiamine produced by community members in the bulk media? Otherwise, this phenomenon is only speculative and should be stated as such.

Response: Thank you for pointing this out. First, about the carbon source of community members, this is a very intriguing topic that has also been discussed by our previous researches (5, 6). THF was the sole carbon source in the microbial

community H-1 at first, but available carbon sources of community members became diversified during THF degradation. This figure may give us a brief understanding of THF degrading pathways predicted by previous work (7-9) (**Fig. R1**). Notice that the degrading of THF creates many intermediates that are consumable, and ZM07 may trade these with the rest of the community for thiamine. We did attempt to detect these intermediates of THF, however, we didn't succeed. We suppose that the following reasons may explain why the intermediates of the first few steps are hard to be detected: (i) the low concentration of intermediates during metabolism make them hard to be detected, (ii) the intermediates of the first few steps are very unstable and may rapidly convert into other intermediate before we can detect them. (iii) the degrading process may happen inside the cell which makes the intermediate difficult to be detected. Additionally, most of the downstream intermediates of THF (such as succinic acid) can also be created in other metabolic pathways, so it is extremely hard to confirm which substances are the carbon sources provided by ZM07. Second, we confirmed that strain ZM07 is a thiamine auxotrophic THF-degrading bacterium through cocultured experiments (strain ZM07 was cocultured with *Escherichia coli* K12 (which cannot degrade THF but can produce thiamine) and/or *Escherichia coli* K12 Δ *thiE* (which can neither degrade THF nor produce thiamine) with or without exogenous thiamine) in our previous research (5). Unfortunately, we are not able to determine this by testing thiamine in the microbial culture H-1 because of its complexity. We are still working on this, and hopefully we can find a more conclusive evidence to demonstrate the exchange between THF intermediate metabolites of the degrader ZM07 and thiamine of the cooperators in our future research. In order to eliminate any misunderstanding, we have added some explanation of this in the manuscript, please check this modification in line 284-287.

Fig. R1 The proposed THF degrading pathways by previous researches (7-9).

3. To emphasize the importance of this research, consider adding perspective how this work can be applied in remediation. How can the findings of this study be applied to field bioremediation of tetrahydrofuran and other xenobiotic compounds? Consider adding this in more detail in the ‘Importance’ section and at the end of the discussion beyond line 294.

Response: Thank you for your insightful comments. We have added some statement on a new strategy to improve the THF or other pollutants biodegradation performance to emphasize the importance of our findings in field bioremediation. Please check these modifications in lines 37-40 and lines 322-324 of the new manuscript.

Specific Comments

1. Line 13: Run-on sentence, better wording is needed.

Response: Thank you for your advice. We have changed “Thiamine as a crucial growth factor, can regulate the functional microbial communities, however, little is known about its effect in bioremediation.” into “As a crucial growth factor, thiamine

can regulate functional microbial communities; however, our current understanding of its effect on bioremediation is lacking.” Please check this modification in lines 13-14 of the new manuscript.

2. Line 21: “the network” - What network? Need to introduce network analysis earlier in abstract.

Response: Thank you very much. We have added a sentence “A microbial cooccurrence network was constructed to help us identify and isolate the cooperators of strain ZM07 in the microbial community.” to explain the purpose of the network analysis. Please check this modification in lines 18-20 of the new manuscript.

3. Line 22: “was proved” - better working needed.

Response: Thank you. We have modified this sentence into “Our results suggested that strain ZM11 is a good cooperator of ZM07, and it might be more competitive than other cooperators (e.g., ZM12) in cocultured systems.” Please check this modification in lines 21-23 of the new manuscript.

4. Lines 23-26: run-on sentence, consider rewording.

Response: Thank you for your advice. We have reworded it as “Additionally, two dominant strains in our microbial culture displayed a ‘seesaw’ pattern, and they showed completely different responses to exogenous thiamine. The growth of the THF degrader ZM07 was spurred by additional thiamine (with an increased relative abundance and significant upregulation of most metabolic pathways), while the growth of the cooperator ZM11 was obviously suppressed under the same circumstances.” Please check this modification in lines 23-27 of the new manuscript, thank you.

5. Line 26: “quite **the** opposite”

Response: We apologize for this typo, and we have added “the” in line 27.

6. Line 47: “**a** thiazole ring and **a** pyrimidine ring.”

Response: We are so sorry for our mistakes, and we have corrected these two typos in the new manuscript as you suggested in line 47.

7. Line 55: “others” - vague, consider revising.

Response: Thank you for pointing that out. We have changed “others” into “other microorganisms” in lines 55-56.

8. Lines 70-71: Consider adding additional description of the specific problems THF pollution is causing.

Response: Thank you for your good advice. The description of the specific problems THF pollution causing are described as follows: “Tetrahydrofuran (THF, a structural analog of dioxane) is a universal solvent widely applied in the chemical and pharmaceutical industries. THF is easily transferred to surface water, groundwater and the atmosphere due to its high water solubility and high vapor pressure (10). According to toxicity tests, THF shows central nervous system toxicity with headache, dizziness, and loss of sense of smell (11). The environmental pollution of THF has become an extremely acute problem considering its wide application, physical properties and toxicity.” Please check this modification in lines 69-74 of the new manuscript.

9. Line 72: Consider introducing your previous work (enrichment of H-1) in-text.

Response: Thank you for your suggestion. We have added detailed introduction for our previous work with microbial culture H-1. Please check this modification in lines 75-79 of the new manuscript.

10. Line 86: microbial culture H-1 **was** transferred.

Response: We are so sorry for our mistakes, and we have changed “were” into “was” in line 92.

11. Line 87: “different THF degradation ability with different growth curve” - needs rewording.

Response: We are sorry for our poor grammar. We have changed this sentence into “The results showed that regardless of the THF degradation ability or growth ability of H-1, it varied in the THI group (THI group includes T1-THI and T40-THI, which represent samples of the 1st and 40th transfers, respectively) and CK group (CK group includes T1-CK and T40-CK, which represent samples of the 1st and 40th transfers, respectively) (Fig. 1).” Please check this modification in lines 92-96 of the new manuscript.

12. Lines 92-95: Lengthy sentence, consider revising.

Response: Thank you for your advice. We have changed this lengthy sentence as follows: Our results suggested that exogenous thiamine could improve the performance of H-1. Compared to the culture under thiamine-poor conditions (CK group), H-1 can degrade THF more efficiently and produce more biomass under thiamine-rich conditions (THI group). The final degradation ratio and biomass of the THI group were 1.48 times ($p < 0.001$) and 2.03 times ($p < 0.001$) those of the CK

group, respectively (**Fig. 1a and b**). Additionally, the final pH value of the THI group was just 0.96 times ($p < 0.001$) that of the CK group (**Fig. 1c**), probably caused by the production and accumulation of higher concentrations of acidic intermediates after degrading higher amounts of THF, as previously reported (6). Please check this modification in lines 100-107 of the new manuscript.

13. Figure 2d: very difficult to distinguish columns between treatment groups (T1-CK, T1-THI, T40-CK, T40-THI). Consider adding spaces between groups. Same for Figure S2.

Response: We are sorry for our unclear figures. We have added spaces between different treatment groups in **Figure 2d** and **Figure S2**.

14. Figure 2e: The similarity squares along the bottom axis are very difficult to interpret. Furthermore, the short explanation in text “In addition, the community heatmap of samples showed that T40-CK is closer to T1-CK than T1-THI” offers little interpretation. Can this relationship be instead proven through statistical analysis? At the very least, the graphic needs to be revised to improve clarity and in-text explanation needs to be expanded. In addition, units should be added to the color spectrum legend.

Response: We are so sorry for our mistake. It cannot be drawn the conclusion from the community heatmap. However, we can come to a similar conclusion through β -diversity index analysis. Therefore, we have deleted the **Fig. 2e** and added some description in lines 125-129 of the new manuscript.

15. Line 115: Unclear what “transfer” is referring to.

Response: We are sorry for our unclear description. “Transfer” means the process we transfer amount of microbial community H-1 to the new culture medium of the next

generation. We have modified it in lines 128-129 of the new manuscript.

16. Line 119: Define DEG the first time it is used in each section.

Response: Thank you for your advice. We have added the definition of DEG when it is first used in each section. Please see it in line 136 and line 387 of the new manuscript. Furthermore, we have also defined DEGs in each caption.

17. Line 120: “A plenty” needs to be reworded.

Response: Thank you very much. We have changed it into “a large number of” in line 138 of the new manuscript.

18. Line 121: “carbon metabolism, ribosome, biosynthesis of amino acids,”-awkward phrasing, needs to be reworded.

Response: Thank you for pointing this out. We are so sorry for our awkward phrasing. We have changed this sentence into “A large number of DEGs related to the pathways of ‘carbon metabolism’, ‘ribosome biosynthesis’, and ‘amino acid biosynthesis’ were upregulated by exogenous thiamine, while DEGs related to the pathways of ‘ABC transporters’, ‘two-component system’, and ‘purine metabolism’ were downregulated in microbial culture H-1 (Fig. S4).” Please check this modification in lines 138-141 of the new manuscript.

19. Line 122: “ABC transporters, two-component system, purine metabolisms”-awkward phrasing, needs to be reworded.

Response: Thank you so much. Please see the response to the above question.

20. Lines 123-126: Awkward transition between paragraphs, consider removing the last sentence in the previous paragraph (Lines 123-125).

Response: Thank you very much. We have removed the last sentence in the previous paragraph.

21. Line 126: “thiamine ~~metabolie~~ metabolism”

Response: We are sorry for our mistake. It has been revised accordingly.

22. Lines 130-131: “was significantly up-regulated after adding exogenous thiamine in ZM07” -reference figure to support statement.

Response: We are sorry for our carelessness. We have added the reference **Table S3** after this statement in line 149 of the new manuscript.

23. Figures S3a and 3a: Volcano plots are difficult to interpret. The difference between treatment and control groups is indistinguishable. Consider adding separate plots for THI and CK groups and explanation in the captions. Otherwise, consider removing altogether. Also, define DEGs in captions.

Response: Thank you for your suggestion. These two volcano plots are irrelevant in this article, so we have deleted them in the new manuscript. Furthermore, we have defined DEGs in captions. Please check it in line 636, line 645, line 672 and **Supplemental Material**.

24. Figures S3b, 3b, and 5c: Neither the legend nor caption identifies which color corresponds to each treatment group (THI or CK). Captions in S3b and 3b need more detail. Also, define DEGs the first time it is used in each caption.

Response: We are sorry for our unclear description. The legends are in the upper right of the figures, and we have added necessary description in each figure caption. Please

check the modifications in **Supplemental Material**. We also have defined DEGs the first time it is used in each caption.

25. Line 140: “The results we have seen show significant differences” awkward phrasing, needs rewording.

Response: We are sorry for our awkward phrasing. We have changed it into “Our results suggested significant differences in microbial gene expression between the THI and CK groups after 40 transfers (**Fig. 4; Fig. S5**).” Please check it in lines 158-159 of the new manuscript.

26. Lines 143 and 146: “significantly regulated by exogenous thiamine” and “significantly up-regulated in ZM07 with exogenous thiamine” - significant by what criteria?

Response: Thank you very much. Here, we used standardized differential genes expression analysis methods of Novogene Bioinformatics Technology Co., Ltd. (Beijing, China). The significance of gene expression was determined using Padj value, and $P_{adj} < 0.05$ was considered as a statistically significant difference.

27. Line 147: “(one possible electron acceptor of Thm)” - needs a citation or wait to mention in discussion.

Response: Thank you for your suggestion. Reference has been added, please see line 166 of the new manuscript. Otherwise, we have also mentioned cytochrome c synthesis genes are upregulated by exogenous thiamine in ZM07 in lines 242-244.

28. Figure 3c: Needs further explanation in caption and in text.

Response: Thank you for your suggestion. We have added the elaborate explanations

in text and in caption. Please check these explanations in lines 287-289 of the text and lines 648-656 of the caption.

29. Lines 150-151: “the situation in other bacteria was opposite” - needs rewording.

Response: Thank you for your suggestion. We have changed “However, the situation in other bacteria was opposite” into “However, other bacteria performed conversely” in line 169 of the new manuscript.

30. Figure S7 - Needs further explanation in caption and in text. Legend needs units if appropriate.

Response: Thank you for your suggestion. We have added the necessary explanations in caption and in text. Additionally, the legend represents the numerical calculation of FPKM based on hierarchical clustering with no units. The calculation method is as follows: firstly, each row of data is normalized based on $\log_2(\text{FPKM}+1)$; and then the data is standardized using zero-mean normalization. Please check the modifications in **Fig. S7 (Fig. S6 now)** and the main text in lines 170-173. Abbreviation: FPKM, fragments per kilobase of transcript per millions fragments sequenced.

31. Lines 203-204: “~~In our previous studies, we examined~~ ~~We obtained~~ a THF-degrading microbial culture containing a thiamine auxotrophic THF-degrading bacterium ~~in our previous studies~~”.

Response: Thank you very much. We have modified this sentence. Please check it in line 226 of the new manuscript.

32. Line 216: Cut “By the way”.

Response: Thank you very much. It has been revised accordingly.

33. Line 219: “cytochrome c synthesis gene expression had a dramatically increase in ZM07” - Also consider adding reference to a figure to support this.

Response: We are sorry for our carelessness. We have added **Table S4** to support this. Please see it in line 243 of the new manuscript, thank you.

34. Line 246: “the heart of all ecosystems, same growth factors” -needs rewording.

Response: Thank you very much. We have changed this sentence into “Microbial communities are the foundation of diverse ecosystems on Earth. However, the maintenance of such important communities may sometimes rely on a few or even a single growth factor, and the influence of the growth factors on the microbial community may vary greatly by habitat.” Please check this modification in lines 269-272 of the new manuscript.

35. Line 247: “Rumen microbes” - needs rewording

Response: Thank you very much. We have changed “Rumen microbes’ studies” into “Studies on rumen microbial communities”. Please check it in line 272 of the new manuscript.

36. Line 254: Consider starting a new paragraph before “Afterwards,”

Response: Thank you for your suggestion. We have started a new paragraph before “Afterwards,”, and added a sentence “The results suggested that H-1 may have different responses to thiamine other than rumen or intestinal microbial communities.” to summarize the previous paragraph. Please check this modification in lines 279-280 of the new manuscript.

37. Line 297: “our earlier ~~period~~ work”

Response: Thank you very much. We have deleted “period” in lines 330-331 of the new manuscript.

38. Line 300: “1 mL cell suspensions ~~were~~”.

Response: We are so sorry for our mistake. We have changed “was” into “were” in line 334 of the new manuscript.

39. Line 301: What chemical form of thiamine was added? Thiamine HCl or thiamine nitrate?

Response: Thank you. We used thiamine (vitamin B1), neither thiamine hydrochloride nor thiamine nitrate. We have added “The chemicals used in this study are listed in **Supplemental Material**”. Please see the detailed information of the chemicals (including THF and thiamine) in **Supplemental Material**.

40. Line 301: “one group ~~was added~~ with 0.01 mM thiamine”.

Response: Thank you very much. We have changed this description into “the group with no exogenous thiamine was the control group (CK group includes T1-CK and T40-CK, which represent samples of the 1st and 40th transfers, respectively), while the other group with 0.01 mM thiamine added treatment was the experimental group (THI group includes T1-THI and T40-THI, which represent samples of the 1st and 40th transfers, respectively).” Please check this modification in lines 335-339 of the new manuscript.

41. Line 303: “and the other one was not” - needs rewording.

Response: Thank you for your comment. Please see the answer to the above question.

42. Line 320-321: “using conventional method.” Citation needed, (e.g., per manufacturer’s instructions).

Response: Thank you for your comment. We have added the information about the kit used. Please check it in lines 382-383 of the new manuscript.

43. Line 331- Description or citation needed for the “gradient dilution technique.”

Response: Thank you for your comment. We have added the description of the “gradient dilution technique” in lines 362-365. Please check this modification in new manuscript.

44. Line 334- “16S rRNA genes were amplified by PCR₇ using the universal bacterial primers 27F₇ and 1492R.”

Response: Thank you very much. We have revised accordingly.

45. Line 344: “1 mL of the cocultured systems”.

Response: Thank you very much. We have added “of” in line 374 of the new manuscript.

46. Line 351: What detector was used with the GC to measure THF?

Response: Thank you. We used a GC-2014C gas chromatograph equipped with a flame ionization detector (FID) and an AOC-20i autoinjector (GC-2014C, Shimadzu, Japan) to assess the THF concentration. We have added this information in lines 393-395 of the new manuscript.

47. Line 353: What wavelength was used to measure optical density?

Response: Thank you very much. The biomass of the samples was monitored by recording the OD₆₀₀ using a UV-3100PC spectrometer (Mapada, China). We have added the wavelength (OD₆₀₀) in lines 396-397 of the new manuscript.

48. Supplementary Materials, Line 1: “SUPPLEMENTARY ME~~Y~~THODS”

Response: Thank you very much. We have corrected this mistake, please check it in **Supplemental Material**.

Reviewer #2 (Comments for the Author):

The manuscript “Thiamine-mediated microbial interaction between auxotrophic *Rhodococcus ruber* ZM07 and prototrophic cooperators in a tetrahydrofuran-degrading microbial community H-1” (control no. Spectrum04541-22) by Hui Huang and colleagues is experimentally sound and provides interesting data.

However, there are several major drawbacks that must be addressed ahead of my agreement on publication in *Microbiology Spectrum*.

First, there are some discrepancies between results and their interpretation. For example, the authors state in the abstract that “Our results indicate....by participating in NADH producing pathway...”. Although there is some evidence for this hypothesis from the metatranscriptome data, this conclusion should be weakened. Then, the wording “...participating in NADH producing pathway...” is misleading. Is thiamine a cofactor here, or do the authors mean that thiamine upregulates genes involved in NADH-producing pathways? Please clarify in the abstract by rewriting.

Response: Thank you very much for insightful suggestion. We are so sorry for our misleading description. It is exactly as you said, the metatranscriptome sequencing data only showed that additional thiamine upregulates genes involved in

NADH-producing pathways. According to these results, we suspected that exogenous thiamine may improve THF degradation efficiency of ZM07 by upregulating genes involved in NADH-producing pathways since THF hydroxylase could use NADH as an electron donor to catalyze the degradation of THF (12). However, with careful consideration, we found that the evidence we have obtained cannot support the above hypotheses. We have deleted this description in abstract to avoid misunderstanding. Meanwhile, we have mentioned this description in discussion in lines 239-240 of the new manuscript.

Second, the results sections on metatranscriptome analysis with respect to conclusions derived from metabolic data needs to be revised regarding its interpretation. A good beginning is made in lines 193-195. Explain why which metabolic changes are hypothesized to reduce or improve the growth of individual strains.

Response: Sorry about the misunderstanding lead by our manuscript and thank you so much for pointing this out. Actually, the dominant strains in this microbial culture (e.g. ZM07, ZM11 and ZM12) were isolated and their genomes were sequenced. Their sequence data are now available in China National GeneBank Sequence Archive (CNSA) of China National GeneBank DataBase (CNGBdb) with project codes CNP0001612 (ZM07), CNP0001636 (ZM11), and CNP0001637 (ZM12), but these results have not yet been published. We referred to the sequence accession numbers in **MATERIALS AND METHODS** in lines 411-412. Therefore, the changes of the individual strains ZM07 and ZM11 can be analyzed through single-cell genome and metatranscriptome sequencing data. To avoid misunderstanding, we have added the following description in the new manuscript: We analyzed the responses of the thiamine auxotrophic THF-degrading bacterium ZM07 and the non-THF-degrading bacterium ZM11 to exogenous thiamine through metatranscriptome sequencing data combined with the genome sequence data of strains ZM07 and ZM11 sequenced earlier. Please check this modification in lines 203-205 of the new manuscript.

Third, thorough check of English writing and spelling is required.

Response: We are so sorry for our poor English writing and spelling. We have sent our manuscript to American Journal Experts (<https://www.aje.com/>) for language editing by a native English speaker. All the spelling, grammar, and word choice errors in the manuscript have been corrected. In addition, the manuscript has been revised by a native English speaker at AJE to ensure clarity and coherence.

Fourth, the manuscript is not well written and is therefore hard to read.

Response: We are sorry for our terrible English writing. We have carefully checked the English writing and spelling, and sent our manuscript to American Journal Experts (<https://www.aje.com/>) for language editing by a native English speaker.

Examples only, but the whole manuscript must be carefully checked and rewritten:

1. The supplementary figures (at least in the PDF) need to be enlarged.

Response: Thank you for your comment. We have enlarged the figures in **Supplemental Material**.

2. The figures should be numbered for clarity.

Response: Thank you for your comment. We have reordered and numbered the figures in new manuscript.

3. Lines 36-37: better “Thiamine may help ZM07 to occupy...”

Response: Thank you very much. We have changed this sentence into “Thiamine may help ZM07 occupy more niches and utilize more resources,” in lines 36-37 of the new

manuscript.

4. Line 214-216: Cited Figure 3c does not show the genes upregulated by thiamine. The genes involved in NADH synthesis, which were found up-regulated, should be mentioned here or elsewhere.

Response: We are sorry for our mistake. We have changed **Figure 3c** into **Table S4** in line 238. We also added the description in lines 239-240 of the new manuscript.

5. Lines 87, 98, 99: all abbreviations such as THI, CK, T40 etc. must be introduced.

Response: Thank you for your comment. We have introduced these abbreviations in **MATERIALS AND METHODS** in lines 335-339. We have added the explanations of these abbreviations to the beginning of the **RESULTS** section in lines 93-96 of the new manuscript.

6. Line 90: do not start sentences with “And”.

Response: Thank you very much. We have deleted all “And” at the beginning of the sentence in line 99, line 112, and line 405.

7. Line 93: what is “more excellent”? Quantify by fold change.

Response: Thank you very much. We are so sorry for our irrigorous description. Here, the THF degradation efficiency and total biomass of H-1 in THI group were compared with CK group. Statistical significance between THI group and CK group was determined using Student’s *t*-test ($n = 3$). The results showed that exogenous thiamine improved the performance of H-1. Our results suggested that exogenous thiamine could improve the performance of H-1. Compared to the culture under thiamine-poor conditions (CK group), H-1 can degrade THF more efficiently and produce more

biomass under thiamine-rich conditions (THI group). The final degradation ratio and biomass of the THI group were 1.48 times ($p < 0.001$) and 2.03 times ($p < 0.001$) those of the CK group, respectively (**Fig. 1a and b**). Additionally, the final pH value of the THI group was just 0.96 times ($p < 0.001$) that of the CK group (**Fig. 1c**), probably caused by the production and accumulation of higher concentrations of acidic intermediates after degrading higher amounts of THF, as previously reported (6). To facilitate understanding, we have modified it in lines 100-107 to make the description more concrete.

8. Compared TO.

Response: We are so sorry for our mistake. It has been revised in line 101.

9. Line 99: "...during transferring". Please rewrite.

Response: Thank you very much. We have rewritten it into "during the process from the 1st transfer to the 40th transfer" in line 111 of the new manuscript.

10. Line 102: α -diversity.

Response: We are sorry for unclear description. According to α -diversity analysis, there was no significant difference in community richness (**Fig. 2a**) and community evenness (**Fig. 2b**) between T1-THI and T1-CK. We have revised it in lines 114-116 of the new manuscript.

11. Line 118: ...many significant alterations in the microbial community: too general statement; composition or metatranscriptome? Which alterations in detail? Start with the data and end with such a (more precisely reworded) conclusion.

Response: Thank you very much for your insightful suggestions. We have adjusted

our inappropriate description. We have deleted “The results revealed that the exogenous thiamine induced many significant alterations in the microbial community”, and added the conclusion, “These results indicated that exogenous thiamine significantly changed the expression of genes involved in cell metabolism, biological transport, and other life activities in microbial community H-1”, after the data description. Please check this modification in lines 141-143 of the new manuscript.

12. Line 119: Define DEG (differentially expressed genes).

Response: Thank you for your suggestion. We have added the definition of DEG when it first used in each section. Please see it in line 136 and line 387 of the new manuscript.

13. Line 137: Consider Figure S5 as a full figure due to its importance.

Response: Thank you so much for your comment. We have moved **Figure S5** to the text as **Fig. 3**.

14. Line 147, 153 and elsewhere: “facilitate” is not appropriate here, better “result in upregulation of genes involved in...”.

Response: Thank you for your suggestion. We have changed “facilitate” into other appropriate description. Please check these modifications in lines 61-63, lines 164-166, lines 174-176, lines 233-234 and lines 313-315 of the new manuscript.

15. Line 156: ...isolation of bacteria related to strain...

Response: Thank you very much. We have changed this sentence to “Network-directed efficient isolation of bacteria related to strain ZM07” in line 177 of the new manuscript.

16. Line 161: Fortunately? Please describe how the two target bacteria were isolated.

Response: Thank you very much. In our previous work, the THF degrader ZM07 was successfully isolated from the enrichment culture H-1 (5). The cooccurrence network was constructed, and the 16S rRNA gene sequence of ZM07 was compared with the gene library to locate the node corresponding to it in the constructed network. Then, we sought out two nodes connected to strain ZM07. According to the result of network analysis, two non-THF-degrading strains ZM11 and ZM12 were isolated using gradient dilution technique on selective culture medium, i.e., the cell suspension of microbial culture H-1 was diluted into 10^{-1} , 10^{-2} , 10^{-3} , 10^{-4} , 10^{-5} , 10^{-6} and 10^{-7} , microbes were then cultured and isolated using surface spread plate method. (**Fig. 4a**). The separation process of these two target bacteria is described in detail in **MATERIALS AND METHODS** in lines 356-369. Additionally, in order to be more rigorous, we have deleted “Fortunately” of the new manuscript.

17. Lines 162-163: There are much more species and strains in these genera. Do you mean “...in the microbiota analysed here?”

Response: Thank you very much. According to the metatranscriptome sequencing data and the genome sequence data, strain ZM11 and ZM12 we sequenced earlier are the only strain found in this microbial community of their genera (*Hydrogenophaga* and *Pigmentiphaga*) just as you said. To avoid misunderstanding, we have deleted the sentence “Moreover, ZM11 and ZM12 are the only strains of the genera *Hydrogenophaga* and *Pigmentiphaga*, respectively”. As answered in the second question, we have added the description in lines 203-205 to explain the way to observe the metabolic changes of single bacterium.

18. Line 165 Strain ZM07 exhibits...; “much better” is not an appropriate qualification

here, please quantify.

Response: We are sorry for inappropriate statement. We have changed this description into “According to the coculture experiments, both strains could help the degrader restore growth and THF-degrading ability; however, ZM11 appears to be a better cooperater of ZM07 (**Fig. 5b**). The coculture system of ZM07 and ZM12 requires almost 132 h to degrade 20 mM THF, while it only takes the coculture system of ZM07 and ZM11 84 h to consume the same amount of THF.” Please check this modification in lines 183-187 of the new manuscript.

19. Line 175, 266: Here, we propose a seesaw model...

Response: We are sorry for our carelessness. We have modified the mistakes in line 195 and line 299 of the new manuscript.

20. Line 215, Fig. 3c does not indicate the genes mentioned in the text.

Response: We are sorry for our mistake. We have changed **Figure 3c** into **Table S4** in line 238 of the new manuscript.

21. Fig. 1: Red and black colours indicate....

Response: Thank you so much for pointing this out. We have revised this figure and figure legend. Please check these modifications in lines 623-625 and **Fig. 1**.

22. Fig. 2: please explain the scale of the heatmap (1e-1 etc.?)

Response: Thank you very much. The value corresponding to the intermediate heatmap is the z-value obtained after standardization treatment for the relative abundance of the species in each row. Thus, the z-value of a sample in a classification is the value obtained by dividing the difference between the relative abundance of samples and the average relative abundance of all samples by the standard deviation

of all samples. Therefore, the legend represents the z-value with no units, the red and blue represent high and low relative abundances, respectively. According to the comment of the reviewer 1, we have deleted the **Fig. 2e** from the new manuscript.

23. ...and many further examples.

Response: Thank you so much. We have thoroughly checked the English writing and spelling, and sent our manuscript to American Journal Experts (<https://www.aje.com/>) for language editing by a native English speaker.

References

1. Oksanen, J., Kindt, R., Legendre, P., O'Hara, B., Stevens, M.H.H., Oksanen, M.J., Suggests, M. 2007. The vegan package. *Community Ecol Package* 10:631-637. <https://doi.org/10.4135/9781412971874.n145>.
2. Csardi, G., Nepusz, T. 2006. The igraph software package for complex network research. *Inter J Complex Syst* 1695:1-9.
3. Harrell, F.E., Dupont, C. 2018. Hmisc: harrell miscellaneous. R package version 4.1-1. <https://CRAN.R-project.org/package=Hmisc>.
4. Bastian, M., Heymann, S., Jacomy, M. 2009. Gephi: an open source software for exploring and manipulating networks. *Proc. Int. AAAI Conf. Web Soc. Media* 3.
5. Huang, H., Qi, M., Liu, Y., Wang, H., Wang, X., Qiu, Y., Lu, Z. 2020. Thiamine-mediated cooperation between auxotrophic *Rhodococcus ruber* ZM07 and *Escherichia coli* K12 drives efficient tetrahydrofuran degradation. *Front Microbiol* 11:3137. <https://doi.org/10.3389/fmicb.2020.594052>.
6. Liu, Z., Huang, H., Qi, M., Wang, X., Adebajo, O.O., Lu, Z. 2019. Metabolite cross-feeding between *Rhodococcus ruber* YYL and *Bacillus cereus* MLY1 in the biodegradation of tetrahydrofuran under pH stress. *Appl Environ Microbiol* 85:e01196-01119. <https://doi.org/10.1128/AEM.01196-19>.
7. Ren, H., Li, H., Wang, H., Huang, H., Lu, Z. 2020. Biodegradation of tetrahydrofuran by the newly isolated filamentous fungus *Pseudallescheria boydii*

- ZM01. Microorganisms 8:1190. <https://doi.org/10.3390/microorganisms8081190>.
8. Skinner, K., Cuiffetti, L., Hyman, M. 2009. Metabolism and cometabolism of cyclic ethers by a filamentous fungus, a *Graphium* sp. Appl Environ Microbiol 75:5514-5522. <https://doi.org/10.1128/AEM.00078-09>.
 9. Tajima, T., Hayashida, N., Matsumura, R., Omura, A., Nakashimada, Y., Kato, J. 2012. Isolation and characterization of tetrahydrofuran-degrading *Rhodococcus aetherivorans* strain M8. Process Biochem. 47:1665-1669. <https://doi.org/10.1016/j.procbio.2011.08.009>.
 10. Iqbal, A., Ahmad, S.A. 2018. Entrainer based economical design and plantwide control study for Tetrahydrofuran/Water separation process. Chem Eng Res Des 130:274-283. <https://doi.org/10.1016/j.cherd.2017.12.031>.
 11. Joshi, D.R., Adhikari, N. 2019. An overview on common organic solvents and their toxicity. J Pharm Res Int 28:1-18.
 12. Thiemer, B., Andreesen, J. R., & Schröder, T. 2003. Cloning and characterization of a gene cluster involved in tetrahydrofuran degradation in *Pseudonocardia* sp. strain K1. Arch. Microbiol. 179:266-277. <https://doi.org/10.1007/s00203-003-0526-7>.

April 10, 2023

Prof. Zhenmei Lu
Zhejiang University
Institute of Microbiology, College of Life Science, Zhejiang University
Hangzhou, Zhejiang 310058
China

Re: Spectrum04541-22R1 (Thiamine-mediated microbial interaction between auxotrophic *Rhodococcus ruber* ZM07 and prototrophic cooperators in a tetrahydrofuran-degrading microbial community H-1)

Dear Prof. Zhenmei Lu:

Your manuscript has been accepted, and I am forwarding it to the ASM Journals Department for publication. You will be notified when your proofs are ready to be viewed.

Sincerely,

Victor Gonzalez
Editor, Microbiology Spectrum
